# Genome-wide mapping of genetic determinants influencing DNA methylation and gene expression in human hippocampus

Herbert Schulz[1], Ann-Kathrin Ruppert[1], Stefan Herms[2,3,4], Christiane Wolf[5,6,7], Nazanin Mirza-Schreiber[5], Oliver Stegle [6], Darina Czamara[5], Andreas J. Forstner[2,3,8], Sugirthan Sivalingam[2], Susanne Schoch[9,10], Susanne Moebus[11], Benno Pütz [5], Axel Hillmer[12,13], Nadine Fricker[2], Hartmut Vatter[14], Bertram Müller-Myhsok[5,15,16], Markus M. Nöthen[2], Albert J. Becker[9], Per Hoffmann[2,3,4,17], Thomas Sander[1] & Sven Cichon[2,3,4,17]

Emerging evidence emphasizes the strong impact of regulatory genomic elements in neurodevelopmental processes and the complex pathways of brain disorders. The present genome-wide quantitative trait loci analyses explore the *cis*-regulatory effects of single-nucleotide polymorphisms (SNPs) on DNA methylation (meQTL) and gene expression (eQTL) in 110 human hippocampal biopsies. We identify *cis*-meQTLs at 14,118 CpG methylation sites and *cis*-eQTLs for 302 3'-mRNA transcripts of 288 genes. Hippocampal *cis*-meQTL-CpGs are enriched in flanking regions of active promoters, CpG island shores, binding sites of the transcription factor CTCF and brain eQTLs. *Cis*-acting SNPs of hippocampal meQTLs and eQTLs significantly overlap schizophrenia-associated SNPs. Correlations of CpG methylation and RNA expression are found for 34 genes. Our comprehensive maps of *cis*-acting hippocampal meQTLs and eQTLs provide a link between disease-associated SNPs and the regulatory genome that will improve the functional interpretation of non-coding genetic variants in the molecular genetic dissection of brain disorders.

[1] Cologne Center for Genomics, University of Cologne, 50931 Cologne, Germany. [2] Department of Genomics, Life & Brain Center, Institute of Human Genetics, University of Bonn, 53127 Bonn, Germany. [3] Department of Biomedicine, University of Basel, 4031 Basel, Switzerland. [4] Institute of Medical Genetics and Pathology, University Hospital Basel, 4031 Basel, Switzerland. [5] Department of Translational Research in Psychiatry, Max Planck Institute of Psychiatry, 80804 Munich, Germany. [6] European Molecular Biology Laboratory, European Bioinformatics Institute, Hinxton, CB10 1SD Cambridge, UK. [7] Department of Psychiatry, Psychosomatics and Psychotherapy, University of Würzburg, 97080 Würzburg, Germany. [8] Department of Psychiatry (UPK), University of Basel, 4012 Basel, Switzerland. [9] Department of Neuropathology, University of Bonn Medical Center, 53127 Bonn, Germany. [10] Department of Epileptology, University of Bonn Medical Center, 53105 Bonn, Germany. [11] Institute of Medical Informatics, Biometry and Epidemiology, University Duisburg-Essen, 45147 Essen, Germany. [12] Genome Institute of Singapore, Singapore 138672, Singapore. [13] TRON-Translational Oncology at the University Medical Center of the Johannes Gutenberg University gGmbH, 55131 Mainz, Germany. [14] Department of Neurosurgery, University of Bonn Medical Center, 53127 Bonn, Germany. [15] Munich Cluster for Systems Neurology (SyNergy), 81377 Munich, Germany. [16] Institute of Translational Medicine, University of Liverpool, Liverpool L69 3BX, UK. [17] Institute of Neuroscience and Medicine (INM-1), Research Center Jülich, 52425 Jülich, Germany. Herbert Schulz, Ann-Kathrin Ruppert and Stefan Herms contributed equally to this work. Albert J. Becker, Per Hoffmann, Thomas Sander and Sven Cichon jointly supervised this work. Correspondence and requests for materials should be addressed to T.S. (email: sandert@uni-koeln.de) or to S.C. (email: sven.cichon@unibas.ch)

Understanding the functional complexity of the human brain is a major challenge for the genetic dissection of common brain disorders. Genome-wide association studies (GWAS) successfully identified a large number of susceptibility loci for these disorders[1,2]. However, the majority of associated single-nucleotide polymorphisms (SNPs) are located in non-coding genomic regions and usually their functional effects remain elusive[3]. Differential spatiotemporal DNA methylation and gene expression play a key role in normal neurodevelopmental processes and the complex and heterogeneous pathogenesis of brain disorders[4,5]. The rapidly evolving multidimensional epigenomic map of the regulatory genome provides important insights into the functional role of non-coding regulatory elements differentiating diverse transcriptional profiles across a variety of tissue- and cell types[6–9]. Genomic DNA sequence variations have been shown to alter the effects of regulatory genomic elements and thereby influence DNA methylation states and gene expression linking the regulatory genome with individual genetic risk-loci[4,9–21].

Recent methodological advances allow a genome-wide screening for allelic quantitative effects of DNA sequence variants on DNA methylation (methylation quantitative trait loci: meQTLs) as well as gene expression (eQTLs). With regard to the tissue and cell type, as well as context-specific effects of meQTLs and eQTLs, clinically and histopathologically well-characterized human brain tissue is of critical importance to generate high-quality DNA methylation and gene expression profiles[8,9]. Several studies have mapped meQTLs and eQTLs in human brain tissue across multiple regions[14–22]. However, the reliability of the available meQTL and eQTL maps from human brain tissue faces several challenges. First, the brain displays remarkable cellular heterogeneity even within distinct brain regions, confounding cell-type-sensitive DNA methylation states and gene expression patterns[18,20,23]. Second, current sample sizes are relatively small resulting in limited statistical power. Third, individual methylome and transcriptome profiles are usually generated from post-mortem brain tissue. Limitations of post-mortem tissue result from cell damages and DNA/RNA degradation through post-mortem ischemia and tissue preservation[8,24]. These challenges may at least partly explain that the overlap and replicability of meQTLs and eQTLs between independent studies is relatively low[16,20,25]. The current state of knowledge warrants a deeper understanding of specific regulatory processes in the brain to gain novel insights into the underlying biological mechanisms of brain-related traits.

Dysfunction of hippocampal–prefrontal interactions has been implicated in a variety of neurological and psychiatric disorders, such as temporal lobe epilepsy (TLE), Alzheimer's disease, schizophrenia, and depression[26,27]. Given the substantial impact of a polygenic component in the etiology of these common brain disorders, it is of special interest to identify those genetic variants that regulate DNA methylation and gene expression in hippocampal tissue. Accordingly, the present study aims to generate a comprehensive map of cis-acting meQTLs and eQTLs in human hippocampal brain tissue by correlating genome-wide SNP genotypes with high-density CpG methylation and gene expression profiles. Therefore, we took advantage of the unique access to fresh-frozen surgically resected hippocampal biopsies from 110 European patients with pharmacoresistant TLE. By annotating meQTLs and eQTLs to the tissue-specific landscape of regulatory genomic elements characterized by the Encyclopedia of DNA Elements (ENCODE)[7] and the NIH Roadmap Epigenomics Consortia[6], we provide deeper insights into the epigenomic regulation of gene expression in human hippocampal tissue. Specifically, the map of hippocampal meQTLs/eQTLs will improve the functional interpretation of SNPs associated with brain disorders.

## Results

**Study design of meQTL and eQTL analyses.** To explore the cis-regulatory effects of SNPs on DNA methylation and gene expression, we performed genome-wide mapping of cis-acting meQTLs/eQTLs and correlations of CpG methylation and mRNA expression in human hippocampal tissue from 110 European TLE patients (Supplementary Table 1). After stringent array and SNP quality control, 536,041 SNPs pruned by linkage disequilibrium (LD, pair-wise $r^2 < 0.8$ within a window of 50 SNPs), 344,106 CpG probes and 15,708 3'-RNA expression probes were included in the QTL study. We performed cis-meQTL/eQTL analyses within a cis-window of ±500 kb between SNP genotypes and quantitative methylation rate ($\beta$-value) of CpGs or 3'-RNA expression levels, using a linear regression model implemented in Matrix eQTL[28]. We corrected for gender, age at surgery as well as the proportion of neuronal cells, and adjusted by principal components for population stratification, batch effects and for hidden confounders (see Methods). For cis-QTL analyses, we chose a false discovery rate (FDR) of 1%. CpGs and 3'-transcripts with at least one significantly associated SNP (FDR of 1%) were considered as meQTL or eQTL, respectively. The study power was sufficient to detect cis-acting hippocampal QTL-SNPs that

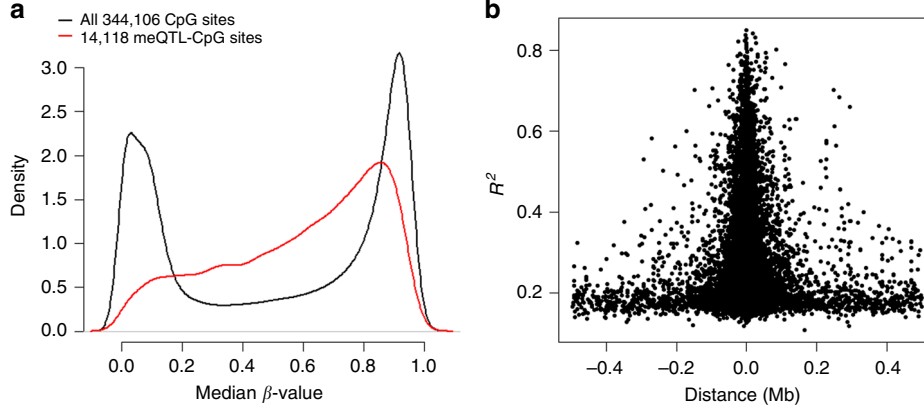

**Fig. 1** DNA methylation in context of genetic determination. **a** Distribution of the degree of methylation ($\beta$-value) of CpGs. The 14,118 meQTL-CpG sites display a rather intermediate distribution of their median $\beta$-values (red) compared to the bimodal distribution observed for all 344,106 CpG sites (black). **b** The relationship of explained variance $R^2$ to the genomic distance of 14,118 meQTL-SNPs and their associated CpG site

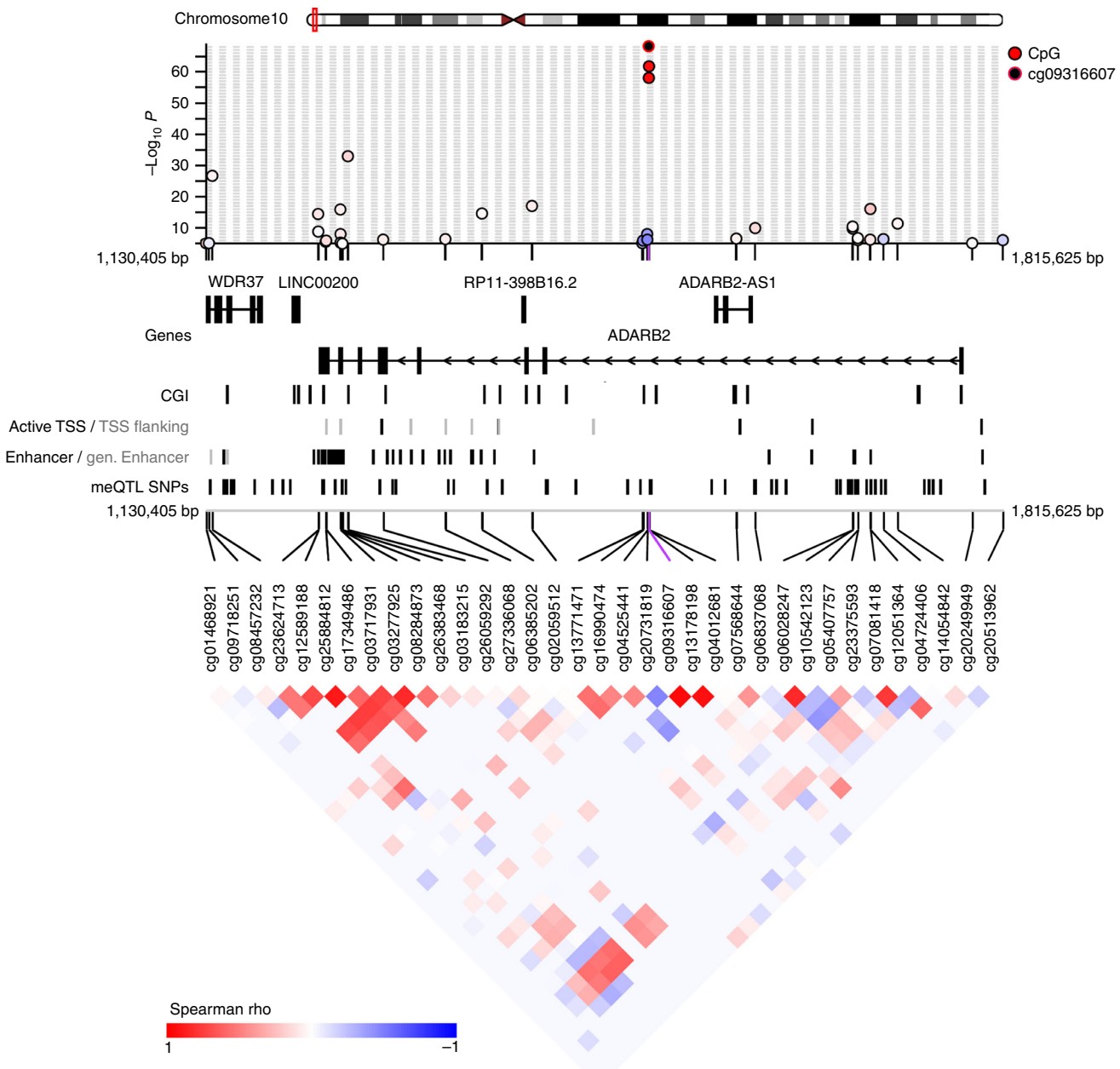

**Fig. 2** Genomic organization of hippocampal *cis*-meQTLs at the *ADARB2* locus. Regional plot of hippocampal *cis*-meQTL *P*-values and CpG co-methylation patterns in the chromosomal region 10p15.3 (chr10:1,130,405–1,815,625, hg19) encompassing the RNA-editing adenosine deaminase-2 gene (*ADARB2*). We display gene symbols (Ensembl), CpG islands (UCSC), the Roadmap Hippocampus middle (E071) 15-chromatin-states: active TSS (#1 TssA, black), flanking active TSS (#2 TssAFlnk, gray), enhancer (#7 Enh, black), genic enhancers (#6 EnhG, gray) and *cis*-meQTL-SNPs. The plot was created using coMET[69]

explained >10% of the variance of CpG methylation, and >16% of the variance in gene expression, respectively.

**Cis-meQTL analyses**. For an FDR of 1% ($P < 9.8 \times 10^{-6}$), *cis*-meQTL analysis identified 66,970 significant SNP-CpG methylation associations at 14,118 CpG sites (Supplementary Data 1). The median *β*-values across all 344k CpGs displayed a bimodal distribution reflecting an excess of either completely methylated or unmethylated CpGs, whereas the hippocampal *cis*-meQTL-CpGs showed a rather unimodal and intermediate distribution with a prominent peak at a *β*-value of 0.86 (Fig. 1a). The median distance between the meQTL-CpGs and the most significantly associated SNP was 11.7 kb (IQR = 3.2–36.7 kb) (Fig. 1b). For

the most significantly associated *cis*-meQTL SNP-CpG pairs, we observed a median *β*-value of 0.65 (IQR = 0.40–0.82) and a median methylation change of 3.4% (Interquartile Range (IQR) = 2.0–5.5%) per allele. SNP genotypic variation explained a substantial proportion of the methylation variance ranging from 10.8 to 84.9% (median = 24.0%; IQR = 19.1–34.2%). Remarkably, the proportion of variance explained by cell-type heterogeneity was modest (~5%, range: 0–54%) and relatively low for the age-at-sampling (~3%, range: 0–39%) and gender (~1%, range: 0–31%), respectively (Supplementary Data 1). Notably, 9375 (66.4%) of *cis*-meQTL-CpGs resided in 4,905 ENSEMBL-73 genes. Overall, 3578 (25.3%) of the hippocampal meQTL-CpGs were located within or nearby 1,140 (73.0%) out of 1561 candidate genes recently implicated for neurodevelopmental disorders

**Table 1 Enrichments of 14,118 *cis*-meQTL-CpGs in regulatory genomic elements**

| Annotation | Ratio meQTL (%) | Ratio non-meQTL (%) | *P*-value | *q*-value | OR (95% CI) |
|---|---|---|---|---|---|
| Roadmap E071 flanking active TSS | 13.2 | 10.1 | $1.4 \times 10^{-27}$ | $2.7 \times 10^{-25}$ | 1.353 (1.282–1.427) |
| Roadmap E071 weak repr. PolyComb | 12.5 | 10.7 | $3.4 \times 10^{-10}$ | $6.5 \times 10^{-08}$ | 1.191 (1.128–1.257) |
| Roadmap E071 H3K27me3 | 49.5 | 46.0 | $2.1 \times 10^{-14}$ | $4.1 \times 10^{-12}$ | 1.148 (1.108–1.189) |
| Roadmap E071 H3K4me1 | 72.1 | 69.1 | $5.5 \times 10^{-13}$ | $1.0 \times 10^{-10}$ | 1.154 (1.109–1.200) |
| Roadmap E071 H3K4me3 | 59.1 | 56.1 | $6.9 \times 10^{-12}$ | $1.3 \times 10^{-09}$ | 1.133 (1.093–1.175) |
| UCSC CGI shore | 31.3 | 27.6 | $4.0 \times 10^{-20}$ | $7.6 \times 10^{-18}$ | 1.198 (1.153–1.245) |
| Brain intermediate methylation regions | 9.8 | 5.6 | $1.3 \times 10^{-74}$ | $2.4 \times 10^{-72}$ | 1.835 (1.724–1.953) |
| Duke DNase I HS cerebellum | 30.9 | 28.4 | $8.4 \times 10^{-10}$ | $1.6 \times 10^{-07}$ | 1.128 (1.085–1.172) |
| Duke DNase I HS cerebrum frontal | 41.7 | 39.7 | $4.6 \times 10^{-06}$ | $8.7 \times 10^{-04}$ | 1.087 (1.049–1.127) |
| Encode (Tfbs) CTCF | 17.3 | 13.6 | $8.3 \times 10^{-31}$ | $1.6 \times 10^{-28}$ | 1.329 (1.267–1.394) |
| Encode (Tfbs) RAD21 | 8.4 | 7.5 | $5.7 \times 10^{-05}$ | $1.1 \times 10^{-02}$ | 1.142 (1.071–1.218) |
| Encode (Tfbs) SMC3 | 5.2 | 4.5 | $1.9 \times 10^{-04}$ | $3.7 \times 10^{-02}$ | 1.167 (1.076–1.265) |
| Encode (Tfbs) POLR2A | 30.8 | 29.2 | $2.1 \times 10^{-04}$ | $4.0 \times 10^{-02}$ | 1.075 (1.035–1.117) |

Co-localization of *cis*-meQTL-CpGs to genomic annotations of 189 regulatory elements were compared with control-CpG sites matched for *β*-value (*n* = 98,826). Significant enrichments after Bonferroni correction (*q*-value) are shown
*OR* odds ratio with 95% confidence interval

based on literature and database queries[29]. Several of these high-ranking candidate genes (e.g., *ADARB2*, *GABRB3/GABRA5*, *HDAC4*, *NRXN1*, *RBFOX3*, *RIMBP2*, *SLC2A1*, and *SLC6A1*) were covered by more than one *cis*-meQTL (Fig. 2; Supplementary Data 1). To facilitate the selection of accessible epigenetic biomarkers, we performed *cis*-meQTL analysis of the 14,118 hippocampal meQTL-CpGs in DNA from whole blood cells of 494 German population controls. We observed a moderate correlation of the estimated meQTL effect sizes between the brain and blood samples (*r* = 0.33). However, about 66% of the hippocampal *cis*-meQTL-CpGs displayed significant SNP-CpG methylation associations (FDR of 1%) with similar effect sizes in whole blood cells (Supplementary Data 1).

**Co-localization of *cis*-meQTLs with regulatory genomic motifs**. To explore the effects of CpG methylation on hippocampal transcription activity[30,31], we examined whether hippocampal *cis*-meQTL-CpGs were co-localized with epigenomic marks annotated by the ENCODE and the Roadmap Epigenomics Projects[6,7]. We interrogated 189 epigenomic marks comprising six hippocampal histone marks and 15 hippocampal chromatin states from Roadmap[6], 161 transcription factor-binding sites (TFBS), three DNAse I hypersensitivity tracks[7,32], three UCSC CpG island (CGI) definitions and regions of intermediate methylation in brain[33]. Given distinct differences of the CpG methylation states across the Roadmap 15-core chromatin states[6] (Supplementary Figs. 1 and 2), we compared the frequency of hippocampal *cis*-meQTL-CpGs within epigenomic marks with those obtained in 98,826 non-meQTL control-CpGs matched for median *β*-values (FDR > 10%, seven-fold match). We found strong relative enrichments of *cis*-meQTL-CpGs in the flanking genomic regions of active promoters (TssAFlnk state, OR = 1.35, *P* = 1.44 × 10^{-27}), marked by an enrichment in chromatin immunoprecipitation sequencing (ChIP-seq) peaks of the promoter-associated histone mark H3K4me3 (OR = 1.13, *P* = 6.85 × 10^{-12}) and the enhancer-associated H3K4me1 mark (OR = 1.15, *P* = 5.51 × 10^{-13}) (Table 1, Supplementary Data 2). Moreover, *cis*-meQTL-CpGs were enriched at chromatin marks with repressed Polycomb states (ReprPCWk state, OR = 1.19, *P* = 3.41 × 10^{-10}) marked by an enrichment in H3K27me3 (OR = 1.15, *P* = 2.14 × 10^{-14}). Otherwise, hippocampal *cis*-meQTL-CpGs were depleted in actively transcribed regions (TX state, OR = 0.60, *P* = 1.81 × 10^{-31}) corresponding with a depletion in H3K36me3 (OR = 0.81, *P*

= 2.96 × 10^{-26}). We observed a strong over-representation of *cis*-meQTL-CpGs in CGI shores (OR = 1.20, *P* = 4.04 × 10^{-20}) and a depletion in neighboring shelves (OR = 0.75, *P* = 3.21 × 10^{-20}). A strong enrichment (OR = 1.84, *P* = 1.26 × 10^{-74}) of *cis*-meQTL-CpGs was found in 6.654 regions of intermediate DNA methylation, which encodes a conserved signature of genome regulation[33]. Notably, hippocampal *cis*-meQTL-CpGs were significantly enriched in the ChIP-seq peaks of the TFBS for CTCF (OR = 1.33, *P* = 8.3 × 10^{-31}) (Table 1), whereas significant depletions were observed for 35 out of 161 ENCODE transcription factors interrogated in this study (Supplementary Data 2).

**Cis-eQTL analyses**. The *cis*-eQTL analysis on quantile normalized expression values was done in a *cis*-window of ±500 kb using the linear regression model of Matrix eQTL. The eQTL analysis led to 1337 significant SNP-3′-RNA expression associations (FDR of 1%, *P* < 4.48 × 10^{-6}) of 302 expression probes of 288 genes (Supplementary Data 3). The median genomic distance was 35.8 kb (IQR = 13.2–76.2 kb) between the SNP and the 5′-TSS of the corresponding gene. The eQTL-SNPs explained proportions of the expression variance ranging from 16.0–79.7% (median = 25.4%; IQR = 20.1–32.3%). Notably, the proportion of variance explained by cell-type heterogeneity was modest (~4%, range: 0–33%) and relatively low for the age-at-sampling (~2%, range: 0–16%) and gender (~1.4%, range: 0–15%), respectively (Supplementary Data 3).

**Correlation of CpG methylation and mRNA expression**. For an FDR of 1%, we detected 80 *cis*-related correlations of CpG methylation and RNA expression, comprising 73 CpGs and 38 mRNA probes, annotating to 34 genes (Supplementary Data 4). The genomic distance between the mRNA transcription start site and CpG site varied between 46 bp and 487.5 kb with a median distance of 11.2 kb (IQR = 0.6–24.6 kb). The CpG methylation levels explained proportions of variance of mRNA expression ranging from 22.3 to 64.2% (median = 27.8%; IQR = 24.7–32.5%). Overall, 70% of the methylation-correlated mRNA expressions were negatively correlated. Particularly, the methylation levels of 30 CpGs in the 5′-regulatory gene regions were predominantly negative correlated with gene expression (OR = 0.267, *P* = 0.025). The majority (50 out of 80) of methylation-correlated mRNA expressions were based on coincidental eQTLs and meQTLs, of

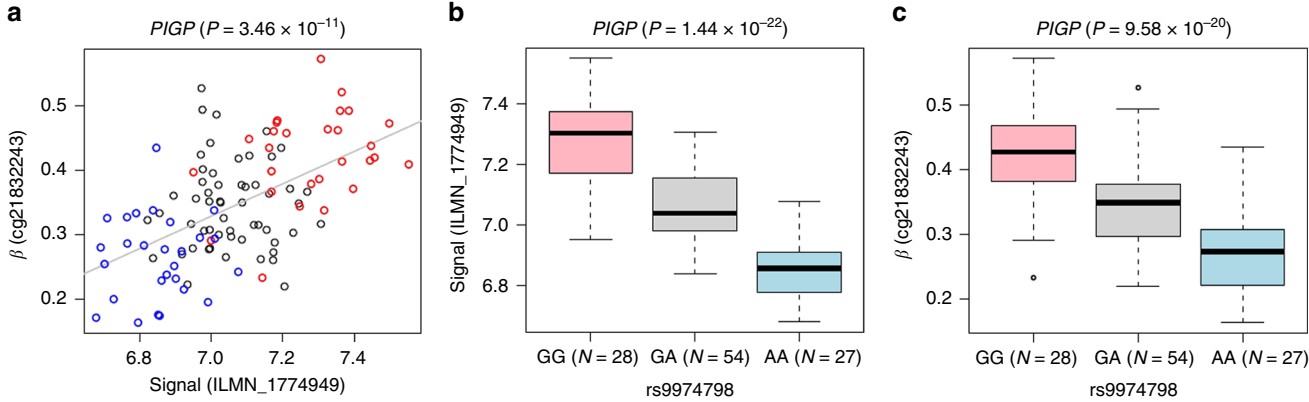

**Fig. 3** Genetically driven correlation of CpG methylation with *PIGP* gene expression. The **a** correlation of *PIGP* 3′-expression with methylation states of CpG cg21832243 in the promoter flanking region (explained variance: 33.5%) is based on **b** a *cis*-eQTL (ILMN_1774949; $P = 1.44 \times 10^{-22}$, explained variance: 57.1%) and **c** a *cis*-meQTL (cg21832243; $P = 9.58 \times 10^{-20}$, explained variance: 50.9%) both driven by a shared SNP rs9974798. Previously, we reported an association of the SNP rs9305614 in the *PIGP* promoter with pharmacoresistance of seizures for the antiepileptic drug levetiracetam in a partial overlapping sample of TLE patients[70]

which 35 pairs of QTL-SNPs were in close LD ($r^2 > 0.4$) indicating a shared genetic driver (Fig. 3).

**LD overlap of *cis*-eQTL- and meQTL-SNPs in brain tissue**. We tested for a clumping-based LD relationship between hippocampal *cis*-meQTL-SNPs with the present set of 1223 hippocampal *cis*-eQTL-SNPs as well as 16,791 *cis*-eQTL-SNPs identified in 10 adult brain regions recently published by Ramasamy and co-workers[18]. Applying the clumping procedure of PLINK[34] (LD pruning of SNPs with pair-wise $r^2 < 0.25$ within 250 kb, maintaining the most significant QTL-SNP), we created a set of 10,890 quasi LD-independent hippocampal *cis*-meQTL-SNPs (FDR of 1%) and a set of quasi LD-independent *cis*-eQTL-SNPs from each eQTL data set (FDR of 1%). For the clumping-based enrichment analysis, we counted the number of 10,890 quasi LD-independent hippocampal *cis*-meQTL-SNPs which showed pair-wise LD ($r^2 > 0.25$ within 1 Mb) with one of the quasi LD-independent *cis*-eQTL-SNPs. Likewise, we randomly generated $10^6$ sets of 10,890 MAF-matched control-SNPs from 92,308 quasi LD-independent non-meQTL control-SNPs (FDR > 10%). Empirical significance for enrichment was determined by counting the number of control-SNP sets that reached or exceeded the number of hippocampal *cis*-meQTL-SNPs showing LD (pair-wise $r^2 > 0.25$ within 1 Mb) with the sets of quasi LD-independent *cis*-eQTL-SNPs. We found significant (empirical $P < 10^{-6}$) relative enrichments of 6.70 of hippocampal *cis*-meQTL-SNPs in the set of hippocampal *cis*-eQTL-SNPs and a 3.62-fold enrichment in the set of brain *cis*-eQTL-SNPs obtained from Ramasamy et al.[18] (Table 2).

**LD overlap of schizophrenia-risk-SNPs with *cis*-QTL-SNPs**. We investigated whether *cis*-acting hippocampal meQTL- and eQTL-SNPs may contribute to 108 risk-loci of schizophrenia recently reported in a large-scale GWAS including 36,989 subjects with schizophrenia and 113,075 population controls[35]. We evaluated two sets of schizophrenia-associated SNPs based on two different significance thresholds of $P < 5 \times 10^{-5}$ and $P < 5 \times 10^{-8}$. We applied the same PLINK clumping procedure described above to assess the number of LD-correlated SNPs between the quasi LD-independent GWAS SNPs and the quasi LD-independent *cis*-meQTL- and *cis*-eQTL-SNPs (Table 3). Empirical significance for enrichment was determined based on $10^6$ randomly generated sets of MAF-matched control-SNPs from the quasi LD-independent non-QTL-SNPs (FDR of 10%, $n = 97,471$). We

found a significant enrichment of *cis*-meQTL-SNPs (relative enrichment: 2.18–2.79) and *cis*-eQTL-SNPs (relative enrichment: 3.60–7.01) with the schizophrenia-associated SNPs (Table 3).

**Epigenomic profiling of hippocampal QTL-SNPs**. To prioritize regulatory, deleterious, and disease-relevant QTL-SNPs among numerous LD-correlated QTL-SNPs, we carried out epigenomic profiling of hippocampal *cis*-meQTL-SNPs ($n = 229,235$ for an FDR of 0.001) and *cis*-eQTL-SNPs ($n = 16,635$ for an FDR of 0.01) derived from the imputed data set of 3,239,626 autosomal SNPs (MAF > 5%). Therefore, we estimated the relative pathogenicity of hippocampal QTL-SNPs using the Combined Annotation Dependent Depletion (CADD v1.3) framework, which integrates annotations across a wide range of functional categories into a single quantitative score (CADD-Phred score) for each SNP[36]. At a threshold of a CADD-Phred score >5, we selected 53,409 meQTL-SNPs at 6226 unique CpG sites (Supplementary Data 5) and 3,899 eQTL-SNPs for 304 unique RNA transcripts (Supplementary Data 6). To facilitate the functional interpretation of the CADD5 QTL-SNPs, we implemented genomic annotations using the Ensemble Variant Effect Predictor (VEP)[37] and hippocampus-related epigenetic annotations for the Roadmap ChromHMM 15-states and relevant histone marks. Notably, 997 CADD5 meQTL-SNPs and 42 CADD5 eQTL-SNPs directly matched GWAS risk-SNPs from the GWASdb v2[38] catalog at a significance level of $P < 5.0 \times 10^{-8}$ (Supplementary Data 5 and 6).

Considering only the QTL-SNPs with the highest CADD score per CpG (best-CADD5 meQTL-SNPs: $n = 6226$; mean CADD score: 12.73, s.d. = 5.32) or RNA transcript (best-CADD5 eQTL-SNPs: $n = 304$; mean CADD score: 14.06, s.d. = 5.19), the distribution of the functional annotations of the CADD top-ranked meQTL-SNPs and eQTL-SNPs displayed a predominant proportion of QTL-SNPs with genic localizations. Remarkably, a considerable proportion of the best-CADD5 meQTL-missense SNPs create/abolish CpG sites and could be a potential target for allele-specific methylation[39,40]. More specifically, we explored potentially regulatory effects of best-CADD5 QTL-SNPs due to their allelic alterations of TFBS affinities predicted by the SNP2TFBS database[41]. In total, 857 out of 6226 best-CADD5 meQTL-SNPs and 44 out of 304 best-CADD5 eQTL-SNPs were predicted to alter TFBSs and the binding affinity of at least one transcription factors (Supplementary Datas 5 and 6). We

**Table 2 Linkage disequilibrium overlap of *cis*-eQTL-SNPs with *cis*-meQTL-SNPs**

| Source | Hippocampal *cis*-eQTL-SNPs | Brain *cis*-eQTLs[a] |
|---|---|---|
| Number of "clumped" *cis*-eQTL-SNPs | 362 | 7961 |
| eQTL-SNPs overlapping meQTL-SNPs | 201/10,890 | 2196/10,890 |
| Mean (eQTL-SNPs overlapping control-SNPs) | 30.0/10,890 | 606.7/10,890 |
| s.d. (eQTL-SNPs overlapping control-SNPs) | 5.09 | 22.29 |
| Error for the mean | 0.010 | 0.044 |
| Average enrichment | 6.70 | 3.62 |
| Empirical *P*-value | $<10^{-6}$ | $<10^{-6}$ |

Overlapping SNPs of the *cis*-eQTL-SNPs and *cis*-meQTL-SNPs were identified by the linkage disequilibrium (LD) clumping procedure implemented in PLINK[34] (SNP pair-wise $r^2 > 0.25$ within 1 Mb). Empirical *P*-values were derived on the basis of 1,000,000 simulated sets of LD-clumped control-SNPs (non-QTL-SNPs with FDR > 10%; LD clumping: SNP pair-wise $r^2 < 0.25$ within 250 kb with reference to the most significantly associated SNP) matched for allele frequency
[a]Brain *cis*-eQTL-SNPs were obtained from a recent exon-level eQTL study from 10 adult brain regions[18]

observed a four-fold enrichment of CTCF-binding sites that overlap with the best-CADD5 meQTL-SNPs ($P = 1.81 \times 10^{-11}$).

## Discussion

Dysfunction of the hippocampal formation and its interaction with the prefrontal cortex plays an important role in neurobiological pathways implicated in a wide variety of human brain disorders[26,27]. The primary aim of our study was to generate genome-wide catalogs of *cis*-regulatory meQTLs and eQTLs in human hippocampal tissue and to explore the functional impact of *cis*-meQTL-CpGs and *cis*-meQTL/eQTL-SNPs on DNA methylation and gene expression. Here we present comprehensive lists of hippocampal *cis*-meQTLs at 14,118 unique CpG sites (Supplementary Data 1) and hippocampal *cis*-eQTLs for 302 3′-RNA probes annotating to 288 genes (Supplementary Data 3). We also provide *cis*-meQTL/eQTL results for the complete set of 3,239,626 imputed SNPs (accessible online: https://uni-bonn. sciebo.de/index.php/s/Nnj2o9GKCmZI2pn). Concomitantly, we carried out epigenomic profiling of hippocampal QTL-SNPs to explore their *cis*-regulatory effects on gene expression and to prioritize pathogenic regulatory SNPs at genetic risk-loci of common brain disorders (Supplementary Data 5 and 6). Besides known rare deleterious coding mutations and structural genomic variations, *cis*-regulatory SNPs affecting hippocampal gene expression may extend the allelic mutation spectrum of brain disorders. Given that rare causal deleterious gene mutations act on a strong polygenic background and the substantial impact of eQTL-SNPs on tissue- and cell type-specific gene expression, compound heterozygosity of rare loss-of-function mutations and common regulatory *cis*-eQTL-SNP alleles lowering gene expression could synergistically aggravate haploinsufficiency of genes causing brain disorders. The compound recessive mode of action may explain the remarkable phenotypic variability among carriers of loss-of-function gene mutations and could account for a relevant proportion of the missing heritability.

The present catalogs of *cis*-acting meQTLs and eQTLs were generated from fresh-frozen hippocampus biopsies, whereas the majority of previous studies used post-mortem specimens. Notably, hippocampus biopsies from TLE patients frequently show neuropathological alterations with neuronal cell loss and gliosis. Thus, it is of critical relevance for QTL analyses to correct for confounding factors such as cell-type heterogeneity and other known or unidentified confounders arising from the TLE-related pathology. To correct for cell-type heterogeneity, we included the individual neuronal proportion[42] in our linear regression model for QTL analysis. For the *cis*-acting hippocampal QTLs (FDR of 1%), the linear regression model revealed a strong impact of SNPs on the variance of CpG methylation and gene expression (average explained variance: 28%), whereas the proportion of variance explained by cell-type heterogeneity was modest (~5%) and

relatively low for the age-at-sampling (~2–3%) and gender (~1%), respectively (Supplementary Datas 1 and 3). With regard to the wide range of effects attributable to these covariates for each probe-set, we provide the estimated proportion of explained variance of each covariate for each single probe-set (Supplementary Data 1 and 3). This information allows distinguishing probes, for which the CpG methylation or gene expression states are strongly influenced by the covariate of interest. We also applied an independent surrogate variable analysis (ISVA)[43]-based adjustment to assess whether our supervised linear model may have missed relevant confounders. The variation covered by the ISVs is mostly represented in at least one of our supervised covariates and principal components (Supplementary Fig. 3). Overall, these analyses confirm the large majority (~88%) of *cis*-acting meQTLs and eQTLs discovered by the supervised linear model, and emphasize that our QTL findings are not adversely affected by hidden confounding factors (Supplementary Data 1 and 3, Supplementary Fig. 3).

The epilepsy pathology underlying our hippocampal specimens may selectively change methylation and transcription levels of some CpGs and mRNAs. Evidence suggests that interactions between genetic variation and environmental factors may contribute to eQTLs and meQTLs[44,45]. However, conditional allele-dependent shifts of mRNA transcription levels by gene-by-environment (GxE) interaction seem to affect only a small fraction (0.4%) of the investigated eQTL-genes and explain relatively small proportions of variance of gene expression[44]. To explore the potential influence of the epilepsy status on the present *cis*-meQTLs/eQTLs, we estimated the proportion of variance of CpG methylation and gene expression attributable to TLE-related clinical factors (number of epileptic seizures, duration of epilepsy, type of antiepileptic medication, and therapy outcome after epilepsy surgery). Compared to the strong impact of SNP genotypes on *cis*-acting hippocampal meQTLs and eQTLs (FDR of 1%; average explained variance: 28%, range: 11–85%), the average proportion of variance of CpG methylation and gene expression explained by the investigated TLE-related factors was relatively small varying between 0.4 and 1.1% (range per probe-set: 0.0–14.9%). Moreover, ISV-adjusted QTL analyses did not provide evidence for a substantial effect of epilepsy-related or environmental factors. The epilepsy status thus appears to exert only marginal effects on CpG methylation and gene expression in the QTLs identified in the present study. However, the epilepsy status or environmental factors may induce an upregulation of the expression levels of at least some genes, possibly even in a genotype-dependent manner, which may increase the power to detect epilepsy trait-related eQTLs. This is supported by a *cis*-eQTL study investigating hippocampus tissue derived from 22 TLE patients and 22 normal individuals, which demonstrated that epilepsy-associated SNPs of an epilepsy GWAS meta-analysis[46] were significantly more enriched with hippocampal *cis*-eQTLs of

**Table 3 Linkage disequilibrium overlap of schizophrenia-associated SNPs with hippocampal *cis*-meQTL-/eQTL-SNPs**

| Hippocampal *cis*-QTL | meQTL | | eQTL | |
|---|---|---|---|---|
| Number of 'clumped' *cis*-QTL-SNPs | 10,890 | | 362 | |
| | | | | |
| GWAS Trait | Schizophrenia | | Schizophrenia | |
| GWAS *P*-value threshold | $P < 5 \times 10^{-5}$ | $P < 5 \times 10^{-8}$ | $P < 5 \times 10^{-5}$ | $P < 5 \times 10^{-8}$ |
| Number of 'clumped' GWAS SNPs | 1806 | 184 | 1806 | 184 |
| | | | | |
| QTL-SNPs overlapping GWAS SNPs | 352 | 51 | 24 | 6 |
| Mean (QTL control-SNPs overlapping GWAS SNPs) | 161.43 | 18.28 | 6.66 | 0.86 |
| s.d. (QTL control-SNPs overlapping GWAS SNPs) | 11.72 | 3.96 | 2.55 | 0.92 |
| Error for the mean | 0.023 | 0.008 | 0.005 | 0.002 |
| Average enrichment | 2.18 | 2.79 | 3.60 | 7.01 |
| Empirical *P*-value | $<10^{-6}$ | $<10^{-6}$ | $<10^{-6}$ | $2.25 \times 10^{-4}$ |

Two significance thresholds were used to select risk-SNPs from a recent large-scale GWAS of schizophrenia[35]. Overlapping SNPs of the *cis*-QTL-SNPs and GWAS SNPs were identified by the linkage disequilibrium (LD) clumping procedure implemented in PLINK[34] (SNP pair-wise $r^2 > 0.25$ within 250 kb). Empirical *P*-values were derived on the basis of 1,000,000 simulated sets of LD-clumped control-SNPs (non-QTL-SNPs with FDR > 10%; LD clumping: SNP pair-wise $r^2 < 0.25$ within 250 kb with reference to the most significantly associated SNP) matched for allele frequency

TLE patients than of normal individuals[47]. In line with this interpretation, we identified a hippocampal *cis*-eQTL for the protocadherin *PCDH7* gene (ILMN_1670383, nominal $P = 6.55 \times 10^{-7}$, $R^2 = 21.1\%$), which is expressed in thalamocortical circuits and the hippocampus. The eQTL-SNP rs7674790 (chr4:31149277, hg19, MAF = 34%) is located in the 3′-region of the *PCDH7* gene and in close LD ($r^2 = 0.90$) with the epilepsy GWAS risk-SNP rs28498976 (chr4:31151357; $P = 5.44 \times 10^{-9}$)[46]. Notably, the *PCDH7* *cis*-eQTL identified in hippocampal biopsies of patients with pharmacoresistant TLE has not been reported in current brain *cis*-eQTL catalogs[9,16,18,47].

Two recent studies reported *cis*-meQTL maps in post-mortem brain specimens, using the HumanMethylation450 array[20,21]. For an FDR of 1%, Jaffe and co-workers[21] identified *cis*-meQTLs at 138,962 unique CpG sites in tissue from prefrontal cortex of 258 adult non-psychiatric individuals. In total, 91.3% (12,894/14,118) of the present hippocampal *cis*-meQTLs were also found in the set of prefrontal cortex *cis*-meQTLs. For a conservative Bonferroni-corrected significance threshold, Hannon et al.[20] reported *cis*-meQTLs at 3243 unique CpG sites in tissue from 166 human fetal brains. Using adult post-mortem tissue from three distinct brain regions (prefrontal cortex, striatum, cerebellum), the majority (83.4%) of fetal brain meQTLs were present in at least one of the investigated adult brain regions and the meQTL effect sizes were highly correlated. For a subset of 1390 fetal brain meQTL-CpGs, which were also investigated in the present study, we observed an overlap of 81.0% with adult hippocampal meQTLs. Taken together, these findings implicate that the majority of hippocampal meQTLs are conserved across adult brain regions and most of them are likely to be developmentally stable. The considerable overlap of brain meQTLs indicates a high replicability and validity of meQTLs in human brain tissue (Supplementary Data 1). In addition, we found that 66.6% of hippocampal *cis*-meQTL-CpGs represent also meQTLs in whole blood cells (Supplementary Data 1). Likewise, we observed a 55% overlap of hippocampal eQTL-genes with corresponding eQTLs in whole blood cells, based on the Genotype-Tissue Expression (GTEx) database[9] (Supplementary Data 3). Obviously, our screening procedure will preferentially detect those QTLs which display strong allelic effects on CpG methylation and gene expression across the bulk of various hippocampal cell-types. Therefore, it is not surprising that many of the *cis*-meQTLs/eQTLs detected in hippocampal bulk tissue are not cell type- or tissue-specific. Future studies in single cells will provide deeper insights into cell-type specificity of QTLs in normal and disease-related brain tissue[48].

Collectively, hippocampal *cis*-meQTL-CpGs were annotated to 4905 ENSEMBL-73 genes and were located within or nearby 1140 out of 1561 candidate genes recently implicated in neurodevelopmental disorders[29]. To deepen insights into basic regulatory mechanisms of the transcriptional activity in the human hippocampus, we examined whether *cis*-meQTL-CpGs were co-localized to 189 functional regulatory epigenetic elements annotated in the adult human hippocampus[6,7,32,33] (Table 1, Supplementary Data 2). We tested the hypothesis that the *cis*-meQTL-CpG itself influences regulatory effects of hippocampal epigenetic marks[30,31]. We found strong enrichment of *cis*-meQTL-CpGs in 6,654 genomic regions characterized by an intermediate methylation state, which has been implicated to encode a conserved epigenomic signature of gene regulation and exon usage[33]. Hippocampal *cis*-meQTL-CpGs were enriched in genomic regions flanking active promoters (TssAFlnk) marked by the promoter-associated histone modification H3K4me3 and the enhancer-associated modification H3K4me1 (Table 1, Supplementary Data 2). Partially deviating from our findings, Hannon and co-workers reported an enrichment of fetal brain *cis*-meQTLs for repressive histone modifications (H3K9me3, H3K27me3) and a depletion for histone modifications associated with active transcription (H3K4me1, H3K36me3)[20]. The deviating chromatin signatures may mainly reflect differences of developmental stages between the fetal brain[20] and adult hippocampus tissue in context with the selection of Roadmap reference data sets fitting to the tissue source and developmental stage. Consistent with results from fetal brain *cis*-meQTLs[20], we observed an enrichment of *cis*-meQTL-CpGs in the ChIP-seq peaks of the TFBS for CTCF, a highly conserved zinc finger protein that acts as an important transcriptional activator by anchoring other transcription factors[13], but also acts as a repressive insulator by blocking enhancer–promoter interactions[49]. The affinity of CTCF-binding motifs has been shown to be sensitive to CpG methylation[30,50]. Altogether, these findings support the hypothesis that *cis*-meQTLs may be involved in CTCF-mediated enhancer–promoter interactions in genomic regions with active chromatin states[13,49].

We identified *cis*-eQTLs for 302 3′-RNA expression probes annotating to 288 genes (Supplementary Data 3). A recent meta-analysis across five eQTL studies of post-mortem human frontal cortex ($n = 424$) reported replicable cortical eQTLs (FDR of 5%) for 158 (55%) out of the 288 hippocampal *cis*-eQTL-genes identified in the present study[16]. Compared with gene-level eQTLs derived from ten brain regions[9], we found an overlap for 193 (67%) out of 288 hippocampal eQTL-genes, but only a moderate replication rate of 89 (31%) out of 288 hippocampal

eQTL-genes when the comparison was restricted to hippocampal tissue (Supplementary Data 4). In addition, 157 (54.5%) out of 288 hippocampal eQTLs were also eQTLs in whole blood cells in the GTEx eQTL database. This substantial overlap implies that many regulatory SNPs exert more ubiquitous effects on gene expression independent of the tissue source. Correlation analysis between *cis*-acting hippocampal CpG methylation and 3′-RNA expression revealed 34 genes with methylation-driven gene expression (Supplementary Data 4). Correlation of CpG methylation and gene expression frequently occurred in coincidence of *cis*-acting hippocampal meQTL and eQTL pairs that often share the same genetic driver SNP (Supplementary Data 4; Fig. 3). Overall, the number of methylation-driven gene expressions identified in our study is remarkably low considering that we have found hippocampal *cis*-meQTL-CpGs at 4905 ENSEMBL-73 genes. Of note, the HumanHT-12 v3 Expression BeadChip employed in this study detects gene-level expression signals. A recent eQTL study of 10 brain regions using the Affymetrix Human Exon 1.0 ST array identified *cis*-eQTLs (FDR of 1%) for 8573 transcript IDs and 21,617 expression IDs, demonstrating that only 29.6–39.2% of the identified *cis*-eQTLs were reflected in a gene-level signal[18]. Accordingly, the number of *cis*-eQTLs and methylation-expression correlations should be much higher at the exon level relative to the gene level. Considering that we have investigated only 1.2% of 28 million CpG sites in the human genome, the identified correlations likely reflect only a small proportion of the real number of methylation-driven gene expressions. To further explore a putative shared genetic control of co-localized *cis*-meQTLs and *cis*-eQTLs, we tested for an LD-based enrichment of the hippocampal *cis*-meQTL-SNPs in the present set of hippocampal *cis*-eQTL-SNPs as well as in exon-level *cis*-eQTL-SNPs identified in 10 adult brain regions published by Ramasamy et al.[18]. We found a 6.70-fold enrichment of the hippocampal *cis*-meQTL-SNPs for the set of hippocampal *cis*-eQTL-SNPs and a 3.62-fold enrichment for the set of exon-level brain *cis*-eQTL-SNPs (Table 2).

To date, the identification of causative SNPs explaining GWAS risk-loci is still a major challenge. To explore the potential impact of *cis*-acting hippocampal meQTLs and eQTLs in common neuropsychiatric disorders, we performed enrichment analyses by comparing an LD relationship of hippocampal *cis*-QTL-SNPs with risk-SNPs identified by a recent large-scale GWAS of schizophrenia[35]. We found a significant 2.2-fold enrichment of hippocampal *cis*-meQTL-SNPs and 3.6-fold enrichment of *cis*-eQTL-SNPs for schizophrenia-associated SNPs[35]. Our findings confirm previous studies demonstrating an enrichment of fetal brain *cis*-meQTL-SNPs and adult brain *cis*-eQTL-SNPs at GWAS risk-loci of schizophrenia and bipolar disorder[20,51,52] and support the hypothesis that the majority of non-coding GWAS risk-SNPs for brain disorders may affect gene expression[3]. The functional consequences of *cis*-acting meQTLs on gene expression and disease susceptibility are a key topic of current research. Emerging evidence suggests that the binding affinity of transcription factors to their genomic binding sites may be influenced by sequence variations as well as the methylation state of CpGs within the core motifs of TFBSs[30,31]. Thus, causative hippocampal meQTL-CpGs and meQTL-/eQTL-SNPs could affect the transcriptional activity of adjacent genes by allelic alterations of TFBSs within brain-specific regulatory elements, such as promoters, enhancers or insulators[10,13].

To dissect causal regulatory QTL-SNPs among LD-correlated QTL-SNPs, we performed epigenomic profiling of *cis*-acting hippocampal meQTL- and eQTL-SNPs (Supplementary Data 5 and 6; Supplementary Fig. 4). We prioritized potentially regulatory hippocampal QTL-SNPs by estimating their pathogenicity with Ensemble VEP tool[37] and CADD-Phred scores[36] in context

of complementary hippocampus-related epigenetic annotations for the Roadmap ChromHMM 15-states and relevant histone marks. We implemented predictions of potentially regulatory effects of QTL-SNPs due to genetic variation of TFBS affinities using the SNP2TFBS database[41]. In total, 857 *cis*-meQTL-SNPs and 44 *cis*-eQTL-SNPs with CADD-scores >5 were predicted to alter TFBSs (Supplementary Datas 5 and 6). Consistent with an enrichment of *cis*-meQTL-CpGs in the ChIP-seq peaks of CTCF, we found a four-fold enrichment of CTCF binding sites overlapping with the best-CADD5 meQTL-SNPs ($P = 1.81 \times 10^{-11}$). A considerable proportion of the best-CADD5 meQTL-SNPs represents deleterious coding SNPs that also create/abolish CpG sites. SNPs affecting CpG sites account for up to 20% of common SNPs in human genome[39,40,53], and were found to be significantly enriched in eQTLs and in trait-associated SNPs[53]. The potentially methylation-sensitive hippocampal QTL-SNPs might exert meaningful susceptibility effects of trait-associated SNPs. Notably, 997 CADD5 meQTL-SNPs and 42 CADD5 eQTL-SNPs directly matched GWAS trait-SNPs from the GWASdb v2 catalog[38] at a significance level of $P < 5.0 \times 10^{-8}$ (Supplementary Data 5 and 6).

In summary, our catalogs of *cis*-acting hippocampal meQTLs and eQTLs provide a valuable resource for the scientific community to identify genetic drivers of epigenetic and transcriptional variation in the human hippocampus and will deepen our insights into neurodevelopmental processes and neurobiological pathways involved in brain disorders. The majority of these QTLs appear to be conserved across brain regions and developmentally stable. More than 50% of hippocampal *cis*-meQTLs and *cis*-eQTLs are also detected in blood cells and could be used as easily accessible epigenetic biomarkers. The regulatory influence of SNPs on hippocampal CpG methylation and gene expression will inform the interpretation of GWASs and epigenome-wide association studies of brain disorders. Epigenomic profiling of hippocampal QTL-SNPs and meQTL-CpGs and their predicted alteration of TFBSs within brain- and cell type-specific promoters and enhancers will facilitate the dissection of causal regulatory SNPs/CpGs at GWAS risk-loci of brain disorders and will provide valuable functional hints for their leading molecular pathways. The currently available catalogs of brain eQTLs and meQTLs are incomplete and emphasize the need for larger sample sizes of specimens from diverse brain regions in context of various neurodevelopmental stages and disease states.

## Methods

**Study participants and surgical specimens**. Biopsies of hippocampal tissue from 110 European patients with chronic pharmacoresistant temporal lobe epilepsy (TLE) were collected in the Epilepsy Surgery Program at Bonn University. All epilepsy patients were medically resistant and underwent surgical removal after standarized presurgical evaluation to achieve seizure control[54]. The clinical parameters of the TLE patients (58 males, 52 females; range of age at seizure onset: 1 to 67 years, average age: 13.1) are summarized in Supplementary Table 1. For each TLE patient, array-based SNP genotyping, gene expression, and methylation profiling were performed in hippocampal brain tissue specimens. Informed and written consent was obtained from all patients. Procedures were carried out in accordance with the Declaration of Helsinki and were approved by the Ethics Committee of the University of Bonn Medical Center (No. 360/12).

Fresh-frozen human hippocampal segments had been surgically removed from identical regions of the hippocampus and were prepared as tissue-slices at cryostat-conditions. All fresh-frozen hippocampal segments were analyzed by an experienced neuropathologist using international standards and a diagnostic classification was established[55,56]. The majority (>65%) of the hippocampal specimens displayed Ammon's horn sclerosis. In a smaller proportion of hippocampal specimens, lesional alterations such as cortical dysplasia or tumors were diagnosed. For the extraction of genomic DNA and RNA, we used up to five tissue sections with a thickness of 20 µm. Isolation of total DNA and RNA was conducted using the AllPrep DNA/RNA Micro Kit (Qiagen, Hilden, Germany) using the manufacturer's recommendations. Quality control of total RNA was monitored by analysis with RNA 6000 nano lab chips on a BioAnalyzer 2100 (Agilent Technologies, Waldbronn, Germany). All used RNA samples showed intact 28S and 18S ribosomal RNA signals and a RNA integrity number of ≥8.

**SNP genotyping and imputation**. SNP genotyping of the 110 genomic DNA samples was performed using the Illumina Human660W SNP array (Illumina, San Diego, CA, USA). SNPs were annotated according to NCBI build 37.2 using the bead pool manifest Human660W-Quad_v1_H. In total, 508k autosomal SNPs were chosen for imputation based on the following quality control metrics: SNP call rate >97%, minor allele frequency (MAF) > 3% and Hardy–Weinberg Equilibrium (HWE) $P > 10^{-6}$. Pre-phasing was performed using the SHAPEITv2 workflow[57]. Imputation was conducted using IMPUTE2[58] using the provided 1000 Genomes haplotypes Phase I integrated variant set release (v3, March 2012)[59] with a MAF > 1%. The post-imputation SNP genotypes were filtered using SNPTESTv2[60] and PLINK 1.9[34]. IMPUTE2 genotype probabilities were converted into best-guess genotype calls. Overall, 3,239,626 imputed autosomal SNPs were selected according to the following inclusion criteria: info quality score >0.90, call rate >99%, MAF > 5%, and Hardy–Weinberg equilibrium $P > 0.001$. To diminish redundancy of associations for the imputed SNP data set, an LD-based SNP pruning was carried out ($r^2 > 0.8$ for a window size of 50 SNPs), resulting in 536,041 SNPs.

**Preparation and filtering of gene expression profiles**. The Illumina HumanHT-12 v3 Expression BeadChip (Illumina, San Diego, CA, USA) was used to assess 3′-mRNA transcription in mRNA samples of 110 hippocampus biopsies. Hybridization of biotin-UTP-labeled cRNA to the expression BeadChips was followed by washing steps as described in the Illumina protocol. The BeadChips were scanned using the Illumina iScan system and RNA expression raw data were quantile normalized on probe level and without background correction using the Expression Module of the GenomeStudio software (v2011.1). The resulting signals were log2 transformed after offset addition (+16). We excluded ambiguous cross-hybridizing expression probes with more than one genomic location according to Illumina and Ensembl v73 gene annotations, and probes containing a SNP with a MAF > 1% according to 1000 Genome phase 1[59] and phase 3[61]. We excluded weakly expressed probes having an Illumina detection $P > 0.05$ in 95% of the samples. After quality filtering, 15,708 expression probes were included in the eQTL analysis.

**Preparation and filtering of DNA methylation data**. Bisulfite conversion of genomic DNA was applied using Zymo EZ DNA Methylation kit (kit #D5001; Zymo Research Corp., Irvine, CA, USA) according to the manufacturer's protocol. A total of 500 ng of bisulfite converted DNA was analyzed using the Illumina Infinium HumanMethylation450 BeadChip (Illumina, San Diego, CA, USA) according to the manufacturer's instructions. The signal intensities of the images were extracted using GenomeStudio version 2011.1 and the HumanMethylation450 manifest version 1 (Illumina, San Diego, CA, USA). Data normalization and filtering was processed using the R-packages "wateRmelon"[62] and "minfi"[63] according to the protocol described by Lehne et al.[64]. In brief, intensities of six probe-type subsets, defined by Infinium assay, color channel and M/U subtype for Infinium type I probes were quantile normalized and intensities with a detection $P ≥ 0.05$ were set missing. To control for technical bias across samples, a PCA over 220 array control probes was performed. According to the distribution of the PC variance proportions, the first six PCs were added as linear predictors to the regression model. Quality filtering of 473,863 autosomal CpG probes was carried out based on the following exclusion criteria: (i) detection $P > 0.01$ in at least 5% of samples ($n = 4333$), (ii) less than three beat counts per probe in at least 5% of the samples ($n = 522$), (iii) cross-hybridization of the CpG probe to more than one genomic location ($n = 29,978$)[65], (iv) SNP (MAF > 1%, 1000 Genome phase 1[59] and phase 3[61] data sets) containing CpG site or 50-mer CpG probes ($n = 107,851$), and (v) CpGs without any SNP (LD-pruned SNP set) within the cis-flanking region of ±500 kb ($n = 96$). After quality filtering, 344,106 CpGs were included in the meQTL analysis.

**Statistical quantitative trait loci analyses**. For cis-meQTL analysis, we confirmed individual sample identity of the SNP and CpG methylation data sets based on the genotypes of 65 CpG/SNP probes on the HumanMethylation450 array compared with the SNP genotype calls assessed by the Illumina Human660W SNP array. The eQTL and meQTL analyses were performed for an additive linear regression model using the R package Matrix eQTL[28]. Given that the majority of hippocampus biopsies showed neuropathological alterations with neuronal cell loss and gliosis, the inter-sample adjustment for the neuronal proportion as well as known and unidentified confounders is of critical relevance for QTL analyses. The neuronal and glial proportion of each hippocampus specimen was calculated using the CETS R package[42]. To design the model, we first build residuals of the expression and methylation matrix (lm, stats R package) over the known confounders: neuronal proportion, age, and gender. Moreover, the first three Eigenstrat[66] PCs from SNP genotype data were included to control for ethnicity differences. For the meQTL model, the six PCs from the 220 array control probes were added to control for technical bias. Subsequent principal component analyses (PCAs) on residuals were calculated using prcomb (stats R package). PCA for RNA expression resulted in a variance proportion of 17.1% and 11.9% for the first two principal components (PCs), respectively. PCA for DNA methylation revealed PCs #1–#3 as an important unknown confounder accounting for 13.5% of variance. These PCs have general impact on RNA expression and DNA methylation,

respectively. Matrix eQTL[28] linear model eQTL analysis was performed using the covariates: gender, age at surgery, neuronal proportion (known confounder), three PCs for population stratification and two additional PCs as unknown confounder. The cis-meQTL analysis was conducted using the covariates: gender, age of surgery, neuronal proportion (known confounder), three PCs for population stratification, six PCs for the adjustment of technical bias and three additional PCs as unknown confounder. In addition, we extracted independent surrogate variables (ISVs) as an alternative method of correction for heterogeneity and latent variables in QTL analyses. We performed a surrogate variable analysis using ISVA[43], gender as the protected variable, fastICA for component extraction and an FDR threshold for feature selection of 5%. Each time 13 ISVs for the methylation and expression matrix, respectively, gender and the three Eigenstrat[66] PCs for correction of ethnic heterogeneity were used for the alternative ISV-adjusted QTL calculations using Matrix eQTL. Cis-QTL analyses were carried out for SNP/probe pairs spanning <500 kb. An FDR threshold of 1% was considered as significant. We did not perform trans-QTL analyses due to the relatively small sample size resulting in an insufficient power and the substantial impact of spurious trans-meQTL associations reflecting cross-hybridization of CpG probes to more than one genomic localization[65,67].

**Correlation analysis of RNA expression and DNA methylation**. Pearson correlation between mRNA expression and CpG methylation states was calculated using Matrix eQTL. Correlation analysis was performed in cis using a window size of ±500 kb around the RNA expression probe. The Pearson coefficients of determination ($R^2$-values) were calculated using R. To prevent side effects from confounding variables correlation was performed on confounder residuals. An FDR threshold of 1% was considered as significant.

**Enrichment analyses of meQTL-CpGs in epigenomic marks**. To explore the effects of CpG methylation on hippocampal transcription activity, we assessed the co-localization of regulatory epigenomic marks and meQTL-CpG sites and compared the number of hits with those obtained in non-meQTL control-CpG sites (FDR of 10%). Given distinct differences of the CpG methylation states across the Roadmap 15-chromatin states[6] (Supplementary Figs. 1 and 2), the hippocampal cis-meQTL-CpGs were matched with the set of control-CpGs for their mean β-values. Therefore, mean β-values of each CpG were determined and assigned to 10 bins (β 0–0.1 to β 0.9–1). According to the subcategory counts for cis-meQTL-CpGs, we performed random sampling without replacement in the control set.

For enrichment analysis, we used the Fisher exact test and selected five sets of epigenomic marks ($n = 189$). We selected broad ChIP-seq peaks of six hippocampus middle histone marks summarized in the Roadmap E071 annotation[6]: H3K4me1, H3K4me3, H3K9me3, H3K36me3, H3K27me3, and H3K27ac. Additionally, the Chromatin states of the Roadmap core 15-state model for hippocampus middle[6], three brain-specific DNase I hypersensitivity tracks from cerebellum, cerebrum frontal, and frontal cortex[7,32], 161 transcription factor ChIP-seq peaks from ENCODE[68], the UCSC CpG islands (CGI) definitions CGI, CGI shore (<2 kb flanking of CGI), and CGI shelf (2–4 kb flanking of CGI) and regions of intermediate DNA methylation[33].

**Enrichment of QTL-SNPs with schizophrenia-associated SNPs**. The enrichment of cis-acting hippocampal meQTL-/eQTL-SNPs (FDR of 1%) in schizophrenia-associated GWAS risk-SNPs[35] (significance thresholds: $P < 5 \times 10^{-5}$ and $P < 5 \times 10^{-8}$) was tested in comparison with MAF-matched control SNPs (FDR > 10%) corresponding to the sets of meQTL-/eQTL-SNPs. Therefore, all SNPs were extracted from the 1000 Genomes Project European SNP data set (phase 3, version 5)[57]. GWAS SNPs associated with schizophrenia were obtained from the Psychiatric Genomics Consortium (PGC). SNPs located in the major histocompatibility complex region (chr6:2,500,000–3,500,000) were excluded from the SNP sets. In a first step, sets of quasi-independent SNPs were separately created for the selected SNP sets, applying the PLINK clumping procedure[34] (pair-wise LD pruning of SNPs ($r^2 < 0.25$ within 250 kb) maintaining the most significant QTL-SNP). Subsequently, we determined the number of LD overlapping ($r^2 > 0.25$ within 250 kb) SNPs between the set of quasi-independent GWAS SNPs and the sets of cis-meQTL-/eQTL-SNPs, using the PLINK clumping procedure. Likewise, we counted the number of overlapping GWAS SNPs for $10^6$ randomly generated sets of MAF-matched quasi LD-independent control SNPs. Empirical significance for enrichment was determined by counting the number of control-SNP sets which reached or exceeded the number of GWAS SNPs overlapping cis-acting hippocampal meQTL-/eQTL-SNPs.

**Enrichment of eQTL variants in hippocampal meQTLs**. The enrichment of hippocampal cis-meQTLs (FDR of 1%) in eQTL-SNPs was assessed in comparison with $10^6$ simulated MAF-matched sets of control-SNPs as described before. Of note, the PLINK clumping procedure for overlapping the hippocampal cis-meQTL-SNPs with cis-eQTL-SNPs was based on an LD relationship of $r^2 > 0.25$ within a window of 1 Mb. Cis-eQTL-SNPs (FDR of 1%) were obtained from the present hippocampal cis-eQTL data set and from a previous cis-eQTL study of 10 human brain regions from 134 healthy European individuals, using the Affymetrix Human Exon 1.0 ST array[18].

**Epigenomic profiling of hippocampal QTL-SNPs**. For the prioritization of potentially regulatory hippocampal *cis*-QTL-SNPs, we selected 229,235 *cis*-meQTL-SNPs with an FDR of 0.001 and 16,635 *cis*-eQTL-SNPs with an FDR of 0.01, which were derived from the QTL results of the imputed data set of 3,239,626 autosomal SNPs (MAF > 5%) to achieve a best possible SNP coverage of the QTL-associated LD blocks. Next, we estimated the relative pathogenicity of hippocampal *cis*-QTL-SNPs using the Combined Annotation Dependent Depletion (CADD v1.3) framework[36]. At a threshold of a CADD-Phred score >5, we selected 53,409 *cis*-meQTL-SNPs at 6226 unique CpG sites and 3899 *cis*-eQTL-SNPs for 304 unique RNA transcripts for functional epigenomic profiling, using genomic annotations from the Ensemble Variant Effect Predictor (VEP)[37], and hippocampus-related (E071) Roadmap Epigenomics[6] annotations for ChromHMM 15-state and histone marks, and the ENCODE[68] transcription factor ChIP-seq peaks of CTCF and POLR2A. Co-localization of hippocampal CADD5 *cis*-QTL-SNPs with TFBSs and allelic alterations of the binding affinities were predicted by the SNP2TFBS database[41]. In addition, we matched overlapping hippocampal CADD5 *cis*-QTL-SNPs with GWAS trait-SNPs from the GWASdb v2 catalog[38] at a significance level of $P < 10^{-5}$.

**Data availability**. Due to data protection issues, the raw data cannot be made publically available. However, individual researchers may request to use the data for specific projects on a collaborative basis. Inquiries should be made to S.C., A.J.B. and T.S. The full results of the eQTL/meQTL study are available at: https://uni-bonn.sciebo.de/index.php/s/Nnj2o9GKCmZI2pn

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

## Acknowledgements

We are grateful to all the patients and control subjects who contributed to this study. We thank Katharina Pernhorst and Natascha Surano for technical support in the sample preparation of hippocampus biopsies. This work was supported by grants from the European Union Seventh Framework Programs (Human Brain Project—HBP, grant 720270 to S.C.; DESIRE, grant 602531 to T.S.; EPITARGET, grant 602102 to A.J.B., S.S., P.H. and S.C.), the German Research Foundation (DFG) within the EUROCORES Program EuroEPINOMICS (grant SA434/5–1 to T.S.), grants of the DFG Research Unit FOR 2715 (to A.J.B. and T.S.), the SFB grant 1089 (to A.J.B. and S.S.), the German Federal Ministry of Education and Research (BMBF) through the Integrated Network IntegraMent (Integrated Understanding of Causes and Mechanisms in Mental Disorders) under the auspices of the e:Med Program (grant 01ZX1314A to M.M.N. and S.C.) and Independent Research Groups in Neuroscience (to S.S.), Fritz-Thyssen Foundation (to A.J.B.), the BONFOR/SciMed program of the University of Bonn Medical Center (A.J.B. and S.S.) and the Koeln Fortune Program of the Faculty of Medicine, University of Cologne (to A.K.R.). M.M.N. is a member of the DFG-funded Excellence-Cluster ImmunoSensation. M.M.N. also received support from the Alfried Krupp von Bohlen und Halbach-Stiftung. Blood samples were drawn from the Heinz Nixdorf Recall Study (HNR) cohort, which is supported by the Heinz Nixdorf Foundation (Germany). Additionally, the study is funded by the German Ministry of Education and Science and the German Research Council (DFG grants: SI236/8–1, SI236/9–1, ER155/6–1; FOR2107, NO246/10–1 to M.M.N.). The genotyping of the Illumina HumanOmni-1 Quad Bead-Chips of the HNR subjects was financed by the German Centre for Neurodegenerative Disorders (DZNE), Bonn.

## Author contributions

A.-K.R., A.J.F., S.Si., S.Sc., S.M., A.H., N.F. and H.V. generated gene expression and methylation data or provided tissue and phenotype data. H.S., A.K.R., S.H., C.W., N.-M.S., O.S., D.C., B.P. and B.M.-M. performed statistical analyses. H.S., A.K.R., P.H., T.S. and S.C. wrote the manuscript. A.J.B., P.H., M.M.N., T.S. and S.C. directed the project and designed the strategies.

## Additional information

**Competing interests:** The authors declare no competing financial interests.

