## [Peer Review File · Nature Communications]

Reviewer #1 (Remarks to the Author):

This is a carefully conducted study on the genetic determinants influencing DNA methylation and mRNA levels in the human hippocampus.

The study has several weaknesses:

- 1) The sample size (n=110) is relatively low when it comes to a QTL analysis. However, the authors validate many of their associations in other brain regions
- 2) All samples derive from people with epilepsy. The disease could have an effect on mRNA and even on DNA methylation.

However, it is reasonable to assume that epilepsy only affects relatively few CpGs.

- 3) The article does not distribute any data which limits the impact and utility of this resource.

However, the study has several strengths including a carefully conducted statistical analysis and a comprehensive annotation analysis which even establishes a connection to schizophrenia.

Also the authors validate some of the reported associations in blood and other brain regions using similar meQTL studies by others.

The literature review seems to be adequate. Overall, the article advances our knowledge and provides useful look up tables.

Major comments

- 1) To increase the impact and utility of the article, make the functional genomic data (gene expression and DNA methylation and relevant sample annotations such as sex and age) publicly available, e.g. Gene Expression Omnibus.

- 2) Less important but desirable. If possible also distribute the SNPs via dbGAP. It would allow others to fit multivariate models relating methylation levels of specific CpGs to SNPs.

- 3) Mention the following limitations in the discussion.

a) low sample size, b) possible bias resulting from epilepsy.

No further comments.

Reviewer #2 (Remarks to the Author):

The authors explore the effects of cis-regulatory SNPs on DNA methylation (meQTL) and gene expression (eQTL) on a genome-wide scale using human hippocampal biopsies. Cis-acting SNPs of hippocampal eQTLs/meQTLs explained on average one fourth of the observed variance of CpG methylation and gene expression whereby cis-meQTLs preferentially mapped to active promoters, CpG island shores, CTCF binding sites, and brain eQTLs. CpG methylation correlated relatively little with gene expression. Importantly, one fourth of the hippocampal cis-meQTLs mapped within or nearby candidate genes implicated in neurodevelopmental disorders, particularly SCZ.

Overall, this study presents an important contribution to the understanding of non-coding DNA variation for DNA methylation and gene expression in human hippocampus, a region well-known to be involved higher brain functions and various diseases, and extends recent studies on this field in a significant way. The results are represented in a coherent and well-organized manner and support overall the conclusions from this interesting work.

The following topics should be addressed by the authors:

Line 129

Hippocampal cis-meQTLs CpGs seem to show a rather intermediate distribution compared with the bimodal distribution of all 344k CpGs. Regions of intermediate methylation have an even higher

likelihood of being associated with distinct cell types. In general, the authors should investigate the potential confounding effects of differential cell type composition in their case/control cohorts beyond the basic neuron/non-neuron confounders. While the latter measure for differential cell type composition is clearly useful, it is unclear to this reviewer how sensitive and quantitative their measure of differential neuron representation is compared to the relatively subtle effect sizes associated with the meQTLs. Can their cell heterogeneity measure indeed resolve differences in neuron composition on the order of 5%? One alternative option is to utilize correction methods devised in the EWAS field (PMID:24464286, for a good overview see PMID:28245214 or PMID:27142380) to try to account for differences in cell type composition.

Line 129

How many SNPs associate with more than one CpG? Such SNPs may encode trans-meQTLs (PMID:27918535). One third of known disease and trait-associated SNPs seem to affect downstream effects on methylation in trans. Do the authors find any transcription factor motifs enriched in 100bp regions around the meQTLs, pointing towards some trans-effects?

Line 149

66% of the hippocampal cis-meQTL CpGs show SNP-CpG association with similar effect sizes in blood cells. Is the methylation state of the corresponding CpGs identical in blood and hippocampus? A scatterplot would be useful here.

To which degree does cross-tissue preservation of hippocampal cis-meQTL CpGs translate into the control of gene expression? Which type of blood cell was analyzed?

Given that hippocampal cis-meQTL CpGs seem to be conserved in blood cells, it would be interesting to know whether they associate with any GWAS risk loci (e.g. cancer) or trait (e.g. height) unrelated to neurodevelopmental disorders such as SCZ.

Line 163

Cis-meQTL CpGs are enriched in active promoters, H3K4me3, H3K4me1, and H3K27me3 and are depleted for actively transcribed promoters (Table 1). To which degree do these different histone marks localize to the same sites and overlap with active (open) promoters that are not transcribed. Authentic Polycomb complexes would strengthen a neurodevelopmental role of cis-meQTL CpGs as has been previously hypothesized by Jaffe and coworkers. Relatedly, CpGs significantly associated with Alzheimer disease (AD) seem also to associate with bivalent domains (PMID:26803900) possibly pointing to a broader role of this functional domain across life span.

Line 172

Cis-meQTL CpGs seem to be enriched in CGI shores. This location has been previously hypothesized to contribute to tissue specific gene expression patterns (PMID:19151715 and PMID:19881528)

Line 183

The authors report rather few cis-eQTLs when compared to previous studies. Does this relate to the disease status and/or technical/analytical issues? This topic should be considered in more detail in the discussion.

Line 215

Large GWA studies are also available for AD – do AD-associated SNPs lack association with hippocampal cis-acting meQTL and eQTL-SNPs supporting a potential neurodevelopmental role or do hippocampal cis-acting meQTL and eQTL-SNPs have a broader role in disease development. Although of smaller size, GWA studies on bipolar disorder appear interesting as well with respect to the genetic architecture of psychiatric disorders.

Line 309

The conclusion that these meQTLs are developmentally stable warrants further evidence. For example, it would very interesting to know, how many of these meQTLs correspond to CpGs that

are developmentally regulated, e.g. by comparing the methylation state of these CpGs between fetal and adult hippocampus or at least between brain tissues. Do they change between cell types or developmental time?

Line 342

Other possibilities for the low correlation between meQTLs and gene expression should be discussed as well. For example, meQTLs may preferentially operate during early development and control gene expression there. Additionally, meQTLs may encode gene expression potential that depends on renewed neuronal activation to manifest. Also consider that biopsies were taken from patients suffering from therapy resistant TLE that may compromise resting gene expression levels.

Line 365

The manuscript does not really support the conclusion that the majority of trait alleles of brain disorders affect the transcriptional regulation of gene expression in a cell-type specific manner. The authors do not perform a cell-type specific analysis and different types of neurons are known to exist in the hippocampus. Moreover, a substantial number of the SNPs associated with DNA methylation and gene expression overlap with those from previous studies using different brain tissues, and inherently, cell types. Similarly, they authors state that 70% of the SNP associated meQTLs seemed to be conserved in blood cells though their effect on gene expression is not reported.

Line 457

The authors should provide further details on the randomization strategy for the array processing and potential batch effects. In particular, the authors should carry out SVA analysis and correlation of individual SVAs with batch, collection date etc. to rule out additional confounders.

In general, I believe that the study would benefit from further exploring the biological relevance of the identified meQTLs. In particular, it would be very interesting to know whether any particular pathways or gene sets are preferentially targeted by these meQTLs. Can the authors provide the results of such enrichment analysis?

Furthermore, a refined discussion could greatly benefit the visibility of this interesting work for a broader readership. Currently, the discussion is fairly lengthy and in large parts redundant with the results section. Instead, the authors could provide further connections and implications with respect to the biological relevance of the observed meQTLs in this section of the paper. In this respect, a graphical abstract and/or flowchart might guide readers less familiar with functional genomics.

Reviewer #3 (Remarks to the Author):

Schulz et al investigate the effect of genetic variants on gene expression and DNA methylation in human hippocampal tissue. The authors use 110 human hippocampal biopsies to detect cis regulatory effects of SNPs yielding 14,118 meQTLs and 302 eQTLs. The authors state that this dataset will provide the basis for a functional interpretation of genetic variants in brain disorders.

The strengths of the paper are the sample size and the availability of fresh frozen tissue. Furthermore, the authors provide an extensive description of the loci found with regard to their biological context.

However, the manuscript may benefit from a few considerations:

Genetic and methylation effects on gene expression. Genetic and epigenetic variation influences

gene expression. Although the separate analysis of eQTLs and meQTLs is useful, I think it would be fantastic to investigate the integrated effect of SNPs and DNAm on gene expression. Can the authors explain more variation in gene expression through the combination of genetic and epigenetic information?

Disease status. The authors state that it is less likely that "meQTLs/eQTLs are not specific to epilepsy considering the marginal impact of genetic factors in the multifactorial etiology of TLE". However, gene expression and DNA methylation can be strongly influenced by disease status or any other environmental factor such as medication etc. How is this accounted for? Could these QTLs be disease specific or medication specific?

Cell type specificity: Gene expression and DNAm profiles are exquisitely cell type specific. The effect of genetic variants is likely to impact different cell types very specifically. There are computational methods available to deconvolute different cell types, at least glial and neuronal cell types as used by the authors. This could be leveraged to show cell type specific eQTLs and meQTLs. As the authors correct for neuronal vs glial cell types, I believe this could be a great extension of the current manuscript.

Gender: Can the authors comment on sex differences?

Power analysis: Can the authors provide a power analysis for both eQTL and meQTL analysis? Are scripts used available online? Are datasets deposited at GEO?

Molecular validation. The authors commented on reasons why current meQTL/eQTL analyses replicate poorly. Given the limitation of the current technology (e.g. interrogation of certain types of genetic variation, tissue specificity etc), it would be helpful to validate the eQTLs and meQTLs using molecular techniques such as CRISPR/Cas through modification of specific SNPs but also DNAm loci. This could validate the presented dataset in particular, as there is no replication sample available.

NCOMMS-17-05407 Response to the Reviewers

We want to thank the reviewers for their thoughtful comments and constructive suggestions, which provide a helpful guidance to improve the quality of this manuscript. In particular, we appreciate the positive feedback from the reviewers, emphasizing the scientific benefit of our catalogs of hippocampal eQTLs and meQTLs for the scientific community. We revised the manuscript according to the critical points raised by the reviewers. We hope that we were able to solve major critical issues and that the revised manuscript will be acceptable for publication in Nature Communications.

Point-by-point response to the comments and suggestions of the reviewers.

(Comments of the reviewers are presented in *ITALICS*)

Reviewer #1 (Remarks to the Author):

This is a carefully conducted study on the genetic determinants influencing DNA methylation and mRNA levels in the human hippocampus.

The study has several weaknesses:

- 1) The sample size (n=110) is relatively low when it comes to a QTL analysis. However, the authors validate many of their associations in other brain regions*
- 2) All samples derive from people with epilepsy. The disease could have an effect on mRNA and even on DNA methylation. However, it is reasonable to assume that epilepsy only affects relatively few CpGs.*
- 3) The article does not distribute any data which limits the impact and utility of this resource.*

However, the study has several strengths including a carefully conducted statistical analysis and a comprehensive annotation analysis which even establishes a connection to schizophrenia. Also the authors validate some of the reported associations in blood and other brain regions using similar meQTL studies by others. The literature review seems to be adequate. Overall, the article advances our knowledge and provides useful look up tables.

Major comments

- 1) To increase the impact and utility of the article, make the functional genomic data (gene expression and DNA methylation and relevant sample annotations such as sex and age) publicly available, e.g. Gene Expression Omnibus.*
- 2) Less important but desirable. If possible also distribute the SNPs via dbGAP. It would allow others to fit multivariate models relating methylation levels of specific CpGs to SNPs.*
- 3) Mention the following limitations in the discussion.*
 - a) low sample size, b) possible bias resulting from epilepsy.*

Point-by-point response to the comments of Reviewer #1:

Ad Point #1: *To increase the impact and utility of the article, make the functional genomic data (gene expression and DNA methylation and relevant sample annotations such as sex and age) publicly available, e.g. Gene Expression Omnibus.*

Ad Point #2: *Less important but desirable. If possible also distribute the SNPs via dbGAP. It would allow others to fit multivariate models relating methylation levels of specific CpGs to SNPs.*

Response: We will make publically available the summary statistics of the hippocampal meQTL/eQTL results (FDR of 1%) in the Supplementary Data 1-6. More in detail, we present the complete summary statistics results of the entire QTL analyses for all imputed 3.2 million SNPs, 344k CpGs and 15k expression probes in the online accessible folder <https://uni-bonn.sciebo.de/index.php/s/Nnj2o9GKCmZI2pn> (content of 26 Gb). We are, however, not in a position to deposit individual array data in publically accessible databases due to restrictions made by our institutional review board. According to the guidelines of the institutional review board, we have to take care for the protection of individual data with respect to the privacy of the study participants (clinical patients affected by pharmaco-resistant epilepsy, 29 out of 110 patients had an age below 18 years). Different to some Anglo-American countries, German institutional review boards appraise the individual interest of privacy more important than the scientific interest of public data access. Consequently, we are not able to make publically available any individual data which could be used as individual identifier. Notably, this restriction also reflects the will of the majority of our study participants who are very motivated to support research but do not want to make publically available their individual datasets in order to ensure data security and to protect privacy. Despite these restrictions with regard to data sharing, it is still possible to get access to the individual data on a collaborative basis. This means that individual researchers are welcome to contact us with a request to use the data for a specific project. We will then set up a data transfer agreement which includes a statement that the data must not be shared by them without our consent. We have implemented this procedure for other data sets with similar restrictions (such as genotype data of psychiatric patients that are included in the Psychiatric Genomics Consortium) and it has worked very well.

Ad Point #3: *Mention the following limitations in the discussion.*

a) low sample size, b) possible bias resulting from epilepsy.

Response:

Ad a) Low sample size: Contrary to reviewer #1, reviewer #3 emphasizes the sample size as an important strength of this study. We do not agree with reviewer #1 that the sample size of 110 hippocampus biopsies is low. For an FDR of 1%, the present sample size was sufficient to identify *cis*-meQTLs at 14,118 CpG sites and *cis*-eQTLs for 302 expression probes based on the strong effects of SNPs on CpG methylation and gene expression (average explained variance: 28%). Notably, the GTEx database presents eQTLs derived from 81 hippocampus samples and a very recent eQTL study published in *Hum Mol Genet* reported hippocampal eQTLs derived from 22 hippocampus biopsies of TLE patients (Mirza et al., *Hum Mol Genet.* 2017 May 1;26(9):1759-1769).

Revision: To provide information on the power of the study cohort, we added the following phrase to the Results/Study design of meQTL and eQTL analyses (“page #4”：“The study power was sufficient to detect *cis*-acting hippocampal QTL-SNPs that explained >10% of the variance of CpG methylation, and >16% of the variance in gene expression respectively.”)

Ad b) Possible bias resulting from the epilepsy status: We appreciate the reviewer’s comment concerning this relevant issue. We agree that CpG methylation and gene expression of some CpGs and genes could be influenced by the epilepsy status and antiepileptic medication compared to hippocampus specimen of healthy subjects. The epilepsy condition may slightly change the level of CpG methylation and gene expression but this shift is genotype-independent and will have only marginal effects on the correlation of methylation/expression with the SNP genotypes.

Revision: We thank the reviewer for the notion that our explanation might be difficult to understand. Therefore we have revised this part of the Discussion (page 9, middle): “The epilepsy pathology underlying our hippocampal specimens may selectively change methylation and transcription levels of some CpGs and mRNAs. However, this general shift should have only marginal effects on the genotypic correlation of SNPs with methylation and expression levels.”

Reviewer #2 (Remarks to the Author):

The authors explore the effects of cis-regulatory SNPs on DNA methylation (meQTL) and gene expression (eQTL) on a genome-wide scale using human hippocampal biopsies. Cis-acting SNPs of hippocampal eQTLs/meQTLs explained on average one fourth of the observed variance of CpG methylation and gene expression whereby cis-meQTLs preferentially mapped to active promoters, CpG island shores, CTCF binding sites, and brain eQTLs. CpG methylation correlated relatively little with gene expression. Importantly, one fourth of the hippocampal cis-meQTLs mapped within or nearby candidate genes implicated in neurodevelopmental disorders, particularly SCZ.

Overall, this study presents an important contribution to the understanding of non-coding DNA variation for DNA methylation and gene expression in human hippocampus, a region well-known to be involved higher brain functions and various diseases, and extends recent studies on this field in a significant way. The results are represented in a coherent and well-organized manner and support overall the conclusions from this interesting work.

The following topics should be addressed by the authors:

Line 129

Hippocampal cis-meQTLs CpGs seem to show a rather intermediate distribution compared with the bimodal distribution of all 344k CpGs. Regions of intermediate methylation have an even higher likelihood of being associated with distinct cell types. In general, the authors should investigate the potential confounding effects of differential cell type composition in their case/control cohorts beyond the basic neuron/non-neuron confounders. While the latter measure for differential cell type composition is clearly useful, it is unclear to this reviewer how sensitive and quantitative their measure of differential neuron representation is compared to the relatively

subtle effect sizes associated with the meQTLs. Can their cell heterogeneity measure indeed resolve differences in neuron composition on the order of 5%? One alternative option is to utilize correction methods devised in the EWAS field (PMID:24464286, for a good overview see PMID:28245214 or PMID:27142380) to try to account for differences in cell type composition.

Line 129

How many SNPs associate with more than one CpG? Such SNPs may encode trans-meQTLs (PMID:27918535). One third of known disease and trait-associated SNPs seem to affect downstream effects on methylation in trans. Do the authors find any transcription factor motifs enriched in 100bp regions around the meQTLs, pointing towards some trans-effects?

Line 149

66% of the hippocampal cis-meQTL CpGs show SNP-CpG association with similar effect sizes in blood cells. Is the methylation state of the corresponding CpGs identical in blood and hippocampus? A scatterplot would be useful here.

To which degree does cross-tissue preservation of hippocampal cis-meQTL CpGs translate into the control of gene expression? Which type of blood cell was analyzed?

Given that hippocampal cis-meQTL CpGs seem to be conserved in blood cells, it would be interesting to know whether they associate with any GWAS risk loci (e.g. cancer) or trait (e.g. height) unrelated to neurodevelopmental disorders such as SCZ.

Line 163

Cis-meQTL CpGs are enriched in active promoters, H3K4me3, H3K4me1, and H3K27me3 and are depleted for actively transcribed promoters (Table 1). To which degree do these different histone marks localize to the same sites and overlap with active (open) promoters that are not transcribed. Authentic Polycomb complexes would strengthen a neurodevelopmental role of cis-meQTL CpGs as has been previously hypothesized by Jaffe and coworkers. Relatedly, CpGs significantly associated with Alzheimer disease (AD) seem also to associate with bivalent domains (PMID:26803900) possibly pointing to a broader role of this functional domain across life span.

Line 172

Cis-meQTL CpGs seem to be enriched in CGI shores. This location has been previously hypothesized to contribute to tissue specific gene expression patterns (PMID:19151715 and PMID:19881528)

Line 183

The authors report rather few cis-eQTLs when compared to previous studies. Does this relate to the disease status and/or technical/analytical issues? This topic should be considered in more detail in the discussion.

Line 215

Large GWA studies are also available for AD – do AD-associated SNPs lack association with hippocampal cis-acting meQTL and eQTL-SNPs supporting a potential neurodevelopmental role or do hippocampal cis-acting meQTL and eQTL-SNPs have a broader role in disease development. Although of smaller size, GWA studies on bipolar disorder appear interesting as well with respect to the genetic architecture of psychiatric disorders.

Line 309

The conclusion that these meQTLs are developmentally stable warrants further evidence. For example, it would very interesting to know, how many of these meQTLs correspond to CpGs that are developmentally regulated, e.g. by comparing the methylation state of these CpGs between fetal and adult hippocampus or at least between brain tissues. Do they change between cell types or developmental time?

Line 342

Other possibilities for the low correlation between meQTLs and gene expression should be discussed as well. For example, meQTLs may preferentially operate during early development and control gene expression there. Additionally, meQTLs may encode gene expression potential that depends on renewed neuronal activation to manifest. Also consider that biopsies were taken from patients suffering from therapy resistant TLE that may compromise resting gene expression levels.

Line 365

The manuscript does not really support the conclusion that the majority of trait alleles of brain disorders affect the transcriptional regulation of gene expression in a cell-type specific manner. The authors do not perform a cell-type specific analysis and different types of neurons are known to exist in the hippocampus. Moreover, a substantial number of the SNPs associated with DNA methylation and gene expression overlap with those from previous studies using different brain tissues, and inherently, cell types. Similarly, they authors state that 70% of the SNP associated meQTLs seemed to be conserved in blood cells though their effect on gene expression is not reported.

Line 457

The authors should provide further details on the randomization strategy for the array processing and potential batch effects. In particular, the authors should carry out SVA analysis and correlation of individual SVAs with batch, collection date etc. to rule out additional confounders.

*In general, I believe that the study would benefit from further exploring the biological relevance of the identified meQTLs. In particular, it would be very interesting to know whether any **particular pathways or gene sets** are preferentially targeted by these meQTLs. Can the authors provide the results of such enrichment analysis?*

*Furthermore, a refined discussion could greatly benefit the visibility of this interesting work for a broader readership. Currently, the discussion is fairly lengthy and in large parts redundant with the results section. Instead, the authors could provide further connections and implications with respect to the **biological relevance of the observed meQTLs** in this section of the paper. In this respect, a graphical abstract and/or flowchart might guide readers less familiar with functional genomics.*

Point-by-point response to the comments and suggestions of Reviewer #2:

General remark: We are grateful for various suggestions of the reviewer, highlighting publications of epigenetic studies for which the present catalogs of hippocampal eQTLs/meQTLs will be of scientific value to dissect genetically-driven epigenetic signatures in brain disorders. Unfortunately, it will be out of the scope of this article to explore all these suggestions more in detail. In fact, we are in contact with Neuropsychiatric Consortia who are eagerly awaiting our hippocampal QTL findings for the interpretation of the current GWAS meta-analyses. Here, we expect follow-up publications including our QTL findings. To demonstrate the potential benefit of our genome-wide catalogs of hippocampal *cis*-acting eQTLs/meQTLs, we presented enrichment analyses for epigenetic regulatory elements in hippocampal cells, and highlighted a few examples for epigenomic profiling of trait-associated regulatory SNPs (rSNPs) in context of: a) prioritization of causal GWAS risk-SNPs, b) compound heterozygosity of rare loss-of-function gene mutations and common eQTL-SNPs in recessive gene models of brain disorders.

Line 129/ Potential confounding effects of cell-type heterogeneity: *Hippocampal cis-meQTLs CpGs seem to show a rather intermediate distribution compared with the bimodal distribution of all 344k CpGs. Regions of intermediate methylation have an even higher likelihood of being associated with distinct cell types. In general, the authors should investigate the potential confounding effects of differential cell type composition in their case/control cohorts beyond the basic neuron/non-neuron confounders. While the latter measure for differential cell type composition is clearly useful, it is unclear to this reviewer how sensitive and quantitative their measure of differential neuron representation is compared to the relatively subtle effect sizes associated with the meQTLs. Can their cell heterogeneity measure indeed resolve differences in neuron composition on the order of 5%? One alternative option is to utilize correction methods devised in the EWAS field (PMID:24464286, for a good overview see PMID:28245214 or PMID:27142380) to try to account for differences in cell type composition.*

Response: The reviewer refers to a “case/control” cohort implicating a EWAS design of our *cis*-QTL study in hippocampus biopsies of 110 patients with TLE. We would like to note that we did not perform EWAS analyses in case/control cohorts for which confounding by cell-type heterogeneity may be of critical relevance due to the relatively small impact of trait-associated effects on differential expression or methylation. Our present QTL analyses investigate the effects of *cis*-acting SNPs for two quantitative trait categories with a large number of targets, specifically CpG methylation (344k CpGs) and gene expression (15k expression probes). In fact, the effect sizes of QTL-SNPs are strong for hippocampal CpG methylation states and gene expression levels (average explained proportion of variance: 28%) compared to the modest proportion of explained variance of 5% for CETS-derived neuronal proportion. Obviously, our study will preferentially detect those QTLs which display strong allelic effects on CpG methylation and gene expression across the bulk of heterogeneous hippocampal cells. Therefore, it is not surprising that a large fraction of the hippocampal meQTLs/eQTLs is not cell type- or tissue-specific and can also be detected in blood cells. Upcoming studies investigating methylation and gene expression in single cells will provide deeper insights into cell-type specificity of QTLs in normal and disease-related brain tissue. Overall, the study power of the present study is sufficient to detect hippocampal *cis*-QTL-SNPs at an FDR of 1% that explain >

10% of the variance of CpG methylation, and > 16% of the variance in gene expression respectively.

We appreciate the suggestion of the reviewer to carry out a reference free method for cell-type correction to ensure that our supervised linear model did not miss any unidentified confounding factors leading to spurious QTL findings. As suggested by the reviewer, we applied an independent surrogate variable (ISV)-adjustment for our QTL statistics. The ISV-adjusted QTL analyses confirmed 58,555 (87.4%) out of 66,970 of the identified meQTL-SNP correlations and 1,186 (88.8%) out of 1,337 of the eQTL-SNP correlations of the supervised QTL analyses, which indicates that the core results of the supervised QTL analyses are robust to potential confounding sources of variation. We also note that the surrogate variables inferred by ISVA are largely captured by the covariates we consider in our supervised model (see new Supplementary Fig. 3). Collectively, the ISV-adjusted QTL results support our supervised QTL analyses and indicate that our findings are not adversely affected by hidden confounding factors.

Revision: We included the ISV-based QTL results in the Supplementary Data 1 & 3 so that it is possible to compare the QTL results of the supervised model and the ISV-adjusted QTL results directly for each CpG and expression probe. Complementarily, we regressed out the explained variance for three major covariates (CETS-derived neuronal proportion, age and gender), which provides detailed information for each CpG methylation and gene expression probe to which extend these three covariates explain the observed variance of CpG methylation and gene expression based on our supervised linear model (see revised Supplementary Data 1 & 3). We included the new Supplementary Fig. 3 demonstrating the strong correlation of the supervised and ISV-adjusted regression model. In addition, we extended the Discussion (page #9, beginning) and the Method section (page #15, Statistical quantitative trait loci analyses) by the ISV-adjusted QTL analyses.

Line 129/trans-QTL analyses: *How many SNPs associate with more than one CpG? Such SNPs may encode trans-meQTLs (PMID:27918535). One third of known disease and trait-associated SNPs seem to affect downstream effects on methylation in trans. Do the authors find any transcription factor motifs enriched in 100bp regions around the meQTLs, pointing towards some trans-effects?*

Response: We would like to point out that the present study was designed to detect *cis*-acting SNPs influencing CpG methylation and gene expression. We did not search for trans-acting QTLs for reasons explained in the Methods Section “Statistical quantitative trait loci analyses”, Page #15, last sentence of the section: “We did not perform trans-QTL analyses with regard to the relatively small sample size resulting in an insufficient power and the substantial impact of spurious trans-meQTL associations reflecting cross-hybridization of CpG probes to more than one genomic localization^{59,61}.”

Revision: None.

Line 149/Comparison of hippocampus and blood meQTLs: *66% of the hippocampal cis-meQTL CpGs show SNP-CpG association with similar effect sizes in blood cells. Is the methylation state of the corresponding CpGs identical in blood and hippocampus? A scatterplot*

would be useful here. To which degree does cross-tissue preservation of hippocampal *cis-meQTL* CpGs translate into the control of gene expression? Which type of blood cell was analyzed? Given that hippocampal *cis-meQTL* CpGs seem to be conserved in blood cells, it would be interesting to know whether they associate with any GWAS risk loci (e.g. cancer) or trait (e.g. height) unrelated to neurodevelopmental disorders such as SCZ.

Response: For the significant hippocampal *cis-meQTL*-CpGs, we present the corresponding *cis-meQTL*-CpG findings in whole blood cells in order to facilitate the selection of accessible epigenetic biomarkers. We did not intend to systematically evaluate the overlap of meQTLs in blood and brain cells, given that QTL analyses of bulk tissue tend to discover preferentially those CpGs which do not exhibit selective tissue- or cell-type specificity. Besides the reported correlation coefficient ($r = 0.33$), a scatterplot would add little information for biomarker selection at the single CpG level. Given that this article focuses on the systematic discovery of meQTLs in hippocampus biopsies and their functional epigenetic annotation, we opt to not extend our downstream analyses to blood meQTLs. The exploration of blood meQTLs for the interpretation of GWAS risk-SNPs in traits unrelated to brain disorders is out of the scope of this article.

Revision: We addressed this issue in the Discussion more specifically (page #10, beginning): “Obviously, our screening procedure will preferentially detect those QTLs which display strong allelic effects on CpG methylation and gene expression across the bulk of various hippocampal cell-types. Therefore, it is not surprising that a prominent fraction of the hippocampal *cis-meQTLs/eQTLs* detected in hippocampal bulk tissue is not selectively cell type- or tissue-specific. Upcoming studies investigating methylation and gene expression in single cells will provide deeper insights into cell-type specificity of QTLs in normal and disease-related brain tissue⁴⁵.”

Line 163/Enrichment of *cis-meQTL*-CpGs to epigenetic regulatory elements: *Cis-meQTL* CpGs are enriched in active promoters, H3K4me3, H3K4me1, and H3K27me3 and are depleted for actively transcribed promoters (Table 1). To which degree do these different histone marks localize to the same sites and overlap with active (open) promoters that are not transcribed. Authentic Polycomb complexes would strengthen a neurodevelopmental role of *cis-meQTL* CpGs as has been previously hypothesized by Jaffe and coworkers. Relatedly, CpGs significantly associated with Alzheimer disease (AD) seem also to associate with bivalent domains (PMID:26803900) possibly pointing to a broader role of this functional domain across life span.

Response: We thank the reviewer for this comment. The intersection of hippocampal *cis-meQTL*-CpGs with Roadmap ChromHMM chromatin states, histone marks and DNase I HS in hippocampal tissue are given in Supplementary Data 1. A systematic exploration of hippocampal meQTLs in context of the functional role of polycomb complexes in aging is out of the scope of this article.

Revision: None

Line 172/*Cis-meQTL* CpGs seem to be enriched in CGI shores: *This location has been previously hypothesized to contribute to tissue specific gene expression patterns (PMID:19151715 and PMID:19881528)*

Response: We appreciate the literature hint of the reviewer. Both publications refer to cancer- or stem cell-related studies without any context to brain disorders. Therefore, we did not follow-up this suggestion of the reviewer which would be out of the scope of this article.

Revision: None

Line 183/relatively low number of hippocampal eQTLs: *The authors report rather few cis-eQTLs when compared to previous studies. Does this relate to the disease status and/or technical/analytical issues? This topic should be considered in more detail in the discussion.*

Response: The relatively low number of *cis*-eQTL genes ($n = 288$) is mainly explained by the array type (Illumina HT-12 v3) employed in this study which interrogates expression signals at the gene-level. After stringent QC, 15,708 autosomal expression probes were included in the present eQTL analyses. With respect to the sample size ($n = 110$) and the content of coding genes interrogated by the Illumina HT-12 v3 array, the number of hippocampal eQTL genes detected in the present study is in the expected range compared to other eQTL studies using comparable gene-level expression arrays.

Revision: This topic has been addressed more in detail in the Discussion in context of the higher resolution of novel exon-level array types or RNA-Seq studies (page #11, beginning).

Line 215/Overlap of hippocampal cis-acting eQTL- and meQTL-SNPs with risk-SNPs obtained in large-scale GWAS meta-analyses of common neuropsychiatric disorders: *Large GWA studies are also available for AD – do AD-associated SNPs lack association with hippocampal cis-acting meQTL and eQTL-SNPs supporting a potential neurodevelopmental role or do hippocampal cis-acting meQTL and eQTL-SNPs have a broader role in disease development. Although of smaller size, GWA studies on bipolar disorder appear interesting as well with respect to the genetic architecture of psychiatric disorders.*

Response: In the present article, we focused on the delineation of hippocampal *cis*-acting meQTL- and eQTL-SNPs and their detailed functional epigenetic annotation, which together provide the rational basis for the prioritization of causal rSNPs at GWAS risk loci. In the Supplementary Data 5 & 6, we now linked hippocampal *cis*-acting meQTL- and eQTL-SNPs with GWAS risk-SNPs derived from the GWASdb v2 catalog at a significance level of $P < 10^{-5}$. We are confident that the overlapping QTL- and GWAS risk-SNPs will be of interest for the scientific community. Currently, we are in contact with various Neuropsychiatric Consortia who want to apply our hippocampal QTL findings for the interpretation of the current GWAS meta-analyses.

Revision: To demonstrate the potential benefit of our genome-wide catalogs of hippocampal *cis*-acting eQTLs/meQTLs, we present enrichment analyses for epigenetic regulatory elements in hippocampal cells, and highlighted a few examples for epigenomic profiling of trait-associated rSNPs in context of: a) prioritization of causal GWAS risk-SNPs (Discussion: page #9), b) compound heterozygosity of rare loss-of-function gene mutations and common eQTL-SNPs in recessive gene models of brain disorders (Discussion: page #8).

Line 309/developmental stability of hippocampal meQTLs: *The conclusion that these meQTLs are developmentally stable warrants further evidence. For example, it would very interesting to know, how many of these meQTLs correspond to CpGs that are developmentally regulated, e.g. by comparing the methylation state of these CpGs between fetal and adult hippocampus or at least between brain tissues. Do they change between cell types or developmental time?*

Response: This conclusion was drawn based on two publications reporting meQTLs in fetal and adult brain tissue (Refs #20-21; Hannon et al., (2016) Nat. Neurosci. 19, 48-54; Jaffe et al., (2016) Nat. Neurosci. 19, 40-47), using the same HM450 methylation arrays and statistical methods as employed in the present study. To demonstrate which of our hippocampal meQTLs are developmentally stable, we provide the corresponding meQTL findings in fetal and adult brain as reported in the published meQTL studies (Refs #20-21) in the revised Supplementary Data 1 and 3. Given that our study and the previous meQTL studies investigated bulk brain tissue, we are not able to differentiate meQTL findings at the cell-type level.

Revision: The revised Supplementary Data 1 and 3 demonstrate an overlap of the present hippocampal meQTLs and meQTL reports in fetal and adult brain tissue of two previous studies (Refs #20-21; Hannon et al., (2016) Nat. Neurosci. 19, 48-54; Jaffe et al., (2016) Nat. Neurosci. 19, 40-47).

Line 342/low correlation between meQTLs and gene expression: *Other possibilities for the low correlation between meQTLs and gene expression should be discussed as well. For example, meQTLs may preferentially operate during early development and control gene expression there. Additionally, meQTLs may encode gene expression potential that depends on renewed neuronal activation to manifest. Also consider that biopsies were taken from patients suffering from therapy resistant TLE that may compromise resting gene expression levels.*

Response: We found only few (n = 80, FDR of 1%) correlations between the CpG methylation and gene expression states. We believe that the main reason for the relatively low number of methylation-driven gene expressions is the low number of hippocampal eQTL-genes identified by the employed gene-level expression arrays.

Revision: At the end of the Discussion we mention that the available catalogs of brain eQTLs and meQTLs are incomplete (page #12): “At present, the available catalogs of brain eQTLs and meQTLs are incomplete and emphasize the need for larger sample sizes of specimens from diverse brain regions in the context of various neurodevelopmental stages and disease states.”

Line 365/cell-type specificity: *The manuscript does not really support the conclusion that the majority of trait alleles of brain disorders affect the transcriptional regulation of gene expression in a cell-type specific manner. The authors do not perform a cell-type specific analysis and different types of neurons are known to exist in the hippocampus. Moreover, a substantial number of the SNPs associated with DNA methylation and gene expression overlap with those from previous studies using different brain tissues, and inherently, cell types. Similarly, they authors state that 70% of the SNP associated meQTLs seemed to be conserved in blood cells though their effect on gene expression is not reported.*

Response: We agree with the reviewer that the present QTL study of hippocampal bulk tissue and the QTL studies of fetal and adult brain tissue do not differentiate brain QTLs at the level of specific cell-types. We did not generate gene expression profiles in blood cells of the healthy population controls. Alternatively, we have matched our hippocampal eQTL genes with the eQTL genes identified in hippocampal tissue by the GTEx Consortium.

Revision: Accordingly, we revised this statement in the Discussion:

“Taken together, our results support the prevailing hypothesis that the majority of common non-coding risk-SNPs identified in GWASs of brain disorders affect the transcriptional regulation of gene expression.”

As requested by the reviewer, we also matched our hippocampal eQTLs with blood eQTLs derived from the GTEx eQTL database to facilitate the selection of potential biomarkers in easily accessible blood cells. The overlap of eQTLs in hippocampal and blood cells are shown in the revised Supplementary Data #3.

Line 457/array processing/batch effects: *The authors should provide further details on the randomization strategy for the array processing and potential batch effects. In particular, the authors should carry out SVA analysis and correlation of individual SVAs with batch, collection date etc. to rule out additional confounders.*

Response: We would like to emphasize that this QTL study is not a case-control based EWAS requiring a randomization strategy for cases and controls. Potential batch effects were carefully removed by inter-sample adjustments of the normalized probe signals as described in the Methods Section “Preparation, normalization and filtering of DNA methylation data”. As suggested by the reviewer, we applied an ISV-adjustment in the QTL statistics to eliminate unknown confounders. ISVs showed a strong correlation with the covariates of our linear model, confirming the validity of the majority (88%) of the reported QTL-findings.

Revision: The QTL results based on an ISV-adjustment were included shown in Supplementary Data 1 & 3.

Pathway/gene enrichment analyses

In general, I believe that the study would benefit from further exploring the biological relevance of the identified meQTLs. In particular, it would be very interesting to know whether any particular pathways or gene sets are preferentially targeted by these meQTLs. Can the authors provide the results of such enrichment analysis?

Response: Unfortunately, our enrichment analyses did not reveal any results of interest.

Results: Gene enrichment analysis on the 288 significant *cis*-eQTL-genes using DAVID resulted in a borderline enrichment of the SMART WD40 repeat domain term SM00320:WD40 (FDR corrected $P = 0.049$, 3.7 fold enrichment). Using Magma, we found a moderate enrichment of *cis*-meQTLs for the Reactome term "Activation of the phototransduction cascade" (corrected P -value = 0.032).

Revision: None. If the reviewer feels these inconclusive results should nevertheless be provided we will be happy to do it.

Graphical abstract and/or flowchart of the strategy for the prioritization of trait-associated rSNPs by epigenetic profiling: *Furthermore, a refined discussion could greatly benefit the visibility of this interesting work for a broader readership. Currently, the discussion is fairly lengthy and in large parts redundant with the results section. Instead, the authors could provide further connections and implications with respect to the biological relevance of the observed meQTLs in this section of the paper. In this respect, a graphical abstract and/or flowchart might guide readers less familiar with functional genomics.*

Response and revision: As suggested by the reviewer, we have included Supplementary Fig. 4, presenting our strategy for the prioritization of trait-associated rSNPs by epigenetic profiling.

Reviewer #3 (Remarks to the Author):

Schulz et al investigate the effect of genetic variants on gene expression and DNA methylation in human hippocampal tissue. The authors use 110 human hippocampal biopsies to detect cis regulatory effects of SNPs yielding 14,118 meQTLs and 302 eQTLs. The authors state that this dataset will provide the basis for a functional interpretation of genetic variants in brain disorders.

The strengths of the paper are the sample size and the availability of fresh frozen tissue. Furthermore, the authors provide an extensive description of the loci found with regard to their biological context.

However, the manuscript may benefit from a few considerations:

Genetic and methylation effects on gene expression. *Genetic and epigenetic variation influences gene expression. Although the separate analysis of eQTLs and meQTLs is useful, I think it would be fantastic to investigate the integrated effect of SNPs and DNAm on gene expression. Can the authors explain more variation in gene expression through the combination of genetic and epigenetic information?*

Disease status. *The authors state that it is less likely that “meQTLs/eQTLs are not specific to epilepsy considering the marginal impact of genetic factors in the multifactorial etiology of TLE“. However, gene expression and DNA methylation can be strongly influenced by disease status or any other environmental factor such as medication etc. How is this accounted for? Could these QTLs be disease specific or medication specific?*

Cell type specificity: *Gene expression and DNAm profiles are exquisitely cell type specific. The effect of genetic variants is likely to impact different cell types very specifically. There are computational methods available to deconvolute different cell types, at least glial and neuronal cell types as used by the authors. This could be leveraged to show cell type specific eQTLs and meQTLs. As the authors correct for neuronal vs glial cell types, I believe this could be a great extension of the current manuscript.*

Gender: *Can the authors comment on sex differences?*

Power analysis: *Can the authors provide a power analysis for both eQTL and meQTL analysis? Are scripts used available online? Are datasets deposited at GEO?*

Molecular validation. *The authors commented on reasons why current meQTL/eQTL analyses replicate poorly. Given the limitation of the current technology (e.g. interrogation of certain types of genetic variation, tissue specificity etc), it would be helpful to validate the eQTLs and meQTLs using molecular techniques such as CRISPR/Cas through modification of specific SNPs but also DNAm loci. This could validate the presented dataset in particular, as there is no replication sample available.*

Point-to-point response to the comments of Reviewer #3:

Genetic and methylation effects on gene expression: *Genetic and epigenetic variation influences gene expression. Although the separate analysis of eQTLs and meQTLs is useful, I think it would be fantastic to investigate the integrated effect of SNPs and DNAm on gene expression. Can the authors explain more variation in gene expression through the combination of genetic and epigenetic information?*

Response: We agree with the reviewer that such an integrated analysis is interesting and had this analysis already included in the submitted manuscript, namely in the Results under Section “Expression quantitative trait methylation (eQTM) analyses” (see Supplementary data 4). Correlation analysis between *cis*-acting hippocampal CpG methylation and 3'-RNA expression revealed 34 genes with methylation-driven gene expression (Supplementary Data 4). Correlation of CpG methylation and gene expression frequently occurred in coincidence of *cis*-acting hippocampal meQTL and eQTL pairs which are often associated with the same SNP (Fig. 3). Figure 3 shows an example of a genetically-driven correlation of CpG methylation and expression of the *PIGP* gene.

Revision: We assume that the reviewer was confused by the wording “Expression quantitative trait methylation (eQTM) analyses”. To improve clarity of this section, we replaced “eQTM” by the expression “*cis*-related CpG methylation and mRNA expression” and revised this section (Results/Correlation analysis of *cis*-related CpG methylation and mRNA expression, page #6, beginning).

Disease status: *The authors state that it is less likely that “meQTLs/eQTLs are not specific to epilepsy considering the marginal impact of genetic factors in the multifactorial etiology of TLE“. However, gene expression and DNA methylation can be strongly influenced by disease status or any other environmental factor such as medication etc. How is this accounted for? Could these QTLs be disease specific or medication specific?*

Response: We appreciate the comment of the reviewer concerning this relevant issue. We agree that CpG methylation and gene expression of some CpGs and genes could be influenced by the epilepsy status, antiepileptic medication and other environmental factors compared methylation/gene expression states in hippocampus specimen of healthy subjects. The epilepsy condition may change the level of CpG methylation and gene expression but this shift should be independent from the individual SNP genotype.

Revision: We thank the reviewer for the hint that our explanation might be difficult to understand. Therefore we have revised the sentence in the Discussion (page 8, middle): “The epilepsy pathology underlying our hippocampal specimens may selectively change methylation and transcription levels of some CpGs and mRNAs. However, this general shift should have only marginal effects on the genotypic correlation of SNPs with methylation and expression levels.”

Cell type specificity: *Gene expression and DNAm profiles are exquisitely cell type specific. The effect of genetic variants is likely to impact different cell types very specifically. There are computational methods available to deconvolute different cell types, at least glial and neuronal cell types as used by the authors. This could be leveraged to show cell type specific eQTLs and*

meQTLs. As the authors correct for neuronal vs glial cell types, I believe this could be a great extension of the current manuscript.

Response: The present QTL study of hippocampal bulk tissue does not differentiate brain QTLs at the level of specific cell-types. Our QTL analyses of hippocampus bulk tissue tend to discover preferentially those CpGs and genes which do not exhibit selective tissue- or cell-type specificity. We thank the reviewer for the suggestion to apply the CETS algorithm to distinguish HM450 methylation profiles for neuronal and glial cells which account for the predominant cell-types in hippocampal tissue. Notably, the CETS algorithm for the transformation of brain bulk tissue into an expected neuronal or glial profile was criticized by a recent publication (PMID:24495553) claiming that the transformation algorithm leads to spurious estimates of neuronal and glial profiles. We are not aware of any software that is able to distinguish neuronal and glial methylation profiles. Therefore, we are not able to provide reliable cell-type specific QTL results proposed by the reviewer. If the reviewer knows a reliable software algorithm for this task, we will be happy to apply it. .

Revision: We used the linear regression model to estimate the proportion of explained variance of CpG methylation and gene expression that is attributable to the CETS-derived cell heterogeneity. For the hippocampal meQTLs/eQTLs, CETS-derived neuronal proportion accounts for approximately 5% of the explained variance of CpG methylation (range: 0% - 54%) and gene expression (range: 0% - 33%). With regard to the wide range of effects attributable to CETS-derived cell heterogeneity across the probe-sets, we extended the Supplementary Data 1 & 3 by additional columns providing the estimated proportion of variance explained by CETS-derived cell heterogeneity for each CpG or mRNA probe-set. This information allows distinguishing probes that are strongly influenced by cell-type composition. Moreover, we included a flag for 414 CpGs in the Supplementary Data 1 (column N, “CETS_Ratio_beta_Neuron_Glia”) that indicates those CpGs (n = 10,000) displaying differential methylation profiles for neuronal and glial cells (PMID:23426267).

Gender: *Can the authors comment on sex differences?*

Response: To assess the impact of the covariates, gender and age, we used our linear regression model to estimate the proportion of explained variance of CpG methylation and gene expression that is attributable to each covariate. For the meQTL-CpGs and eQTL-genes, we observed a relative low impact of gender and age-at-sampling on CpG methylation and gene expression in the 110 hippocampal specimen. Gender-related effects explained approximately 1% (range: 0% - 31%) and age-at-sampling accounted for about 2-3% (range: 0% - 54%) of the variance of CpG methylation and gene expression.

Revision: With regard to the wide range of effects attributable to these covariates for each probe-set, we extended the Supplementary Data 1 & 3 to provide for each probe-set the estimated proportion of explained variance of each covariate. This information allows distinguishing probes, for which the CpG methylation or gene expression states are strongly influenced by the covariate of interest.

Power analysis, Scripts, Data access: *Can the authors provide a power analysis for both eQTL and meQTL analysis? Are scripts used available online? Are datasets deposited at GEO?*

Response: Power estimates can be easily derived from the QTL results. For an FDR of 1%, the present sample size (n = 110) was sufficient to identify *cis*-meQTLs at 14118 CpG sites and *cis*-eQTLs for 302 expression probes based on the strong effects of SNPs on CpG methylation and gene expression (average explained variance: 28%). Overall, the study power was sufficient to detect hippocampal *cis*-QTL-SNPs at an FDR of 1% that explain > 10% of the variance of CpG methylation, and > 16% of the variance in gene expression respectively.

Revision: To provide information about the power of the study cohort, we added the following sentence in the Results/Study design of meQTL and eQTL analyses (“page #4”:“The study power was sufficient to detect *cis*-acting hippocampal QTL-SNPs that explained > 10% of the variance of CpG methylation, and > 16% of the variance in gene expression respectively.”)

Scripts used for data analyses will be provided on request.

Data access: We will make publically available the summary statistics of the hippocampal meQTL/eQTL results (FDR of 1%) in the Supplementary Data 1-6. More in detail, we present the complete summary statistics results of the entire QTL analyses for all imputed 3.2 million SNPs, 344k CpGs and 15k expression probes in the online accessible folder <https://uni-bonn.sciebo.de/index.php/s/Nnj2o9GKCMZI2pn> (content of 26 Gb). We are, however, not in a position to deposit individual array data in publically accessible databases due to restrictions made by our institutional review board. According to the guidelines of the institutional review board, we have to take care for the protection of individual data with respect to the privacy of the study participants (clinical patients affected by pharmaco-resistant epilepsy, 29 out of 110 patients had an age below 18 years). Different to some Anglo-American countries, German institutional review boards appraise the individual interest of privacy more important than the scientific interest of public data access. Consequently, we are not able to make publically available any individual data which could be used as individual identifier. Notably, this restriction also reflects the will of the majority of our study participants who are very motivated to support research but do not want to make publically available their individual datasets in order to ensure data security and to protect privacy. Despite these restrictions with regard to data sharing, it is still possible to get access to the individual data on a collaborative basis. This means that individual researchers are welcome to contact us with a request to use the data for a specific project. We will then set up a data transfer agreement which includes a statement that the data must not be shared by them without our consent. We have implemented this procedure for other data sets with similar restrictions (such as genotype data of psychiatric patients that are included in the Psychiatric Genomics Consortium) and it has worked very well.

Molecular validation: *The authors commented on reasons why current meQTL/eQTL analyses replicate poorly. Given the limitation of the current technology (e.g. interrogation of certain types of genetic variation, tissue specificity etc), it would be helpful to validate the eQTLs and meQTLs using molecular techniques such as CRISPR/Cas through modification of specific SNPs but also DNAm loci. This could validate the presented dataset in particular, as there is no replication sample available.*

Response: Advances in technology and bioinformatic & statistical analyses have improved the accuracy of methylation and gene expression profiling. With regard to the strong effects of SNPs and CpG methylation and gene expression, current QTL studies show reasonable replication rates for meQTLs and also eQTLs. Experimental validation of a larger number of hippocampal meQTLs and eQTLs by epigenomic editing technologies is a very elaborate process and will be

carried out in the context of subsequent studies. In the present study, we used current brain QTL studies to provide confirmatory evidence for our hippocampal meQTLs/eQTLs.

Revision: meQTLs: We used two current meQTL studies of brain tissue that also applied the same HM450 methylation array. We matched overlapping meQTLs to gain confirmatory evidence for replicable brain meQTLs (see revised Suppl. Data 1).

eQTLs: We matched eQTL-genes of three current eQTL studies using independent brain tissue samples (GTEx, Mirza et al., 2017, Kim et al., 2014) to gain confirmatory evidence for replicable brain eQTLs (see revised Suppl. Data 3).

Reviewer #1 (Remarks to the Author):

The authors successfully addressed my comments.

Reviewer #2 (Remarks to the Author):

The revision represents a significant improvement on the original submission in terms of both clarity and quality. Most of my previous concerns are well addressed.

Reviewer #3 (Remarks to the Author):

This is a revised manuscript by Schulz et al on meQTLs and eQTLs in human brain tissue.

I'd like to briefly comment on a remark to reviewer #1, point #3

What I meant with a "good" sample size is the fact that human brain tissue is typically not available in larger N's, thus the sample size from this standpoint is good. My opinion is thus not in contrast to reviewer #1's comment on samples sizes for QTL studies

eQTM analysis: Thank you for clarifying this point, however, I was rather interested on a genome wide model that takes genotype and methylation in relation to gene expression into account. Is the DNAm ~ gene expression relationship for the 34 genes based on previously identified mQTL genes? i.e. are the 73 cis related CpG also found in the mQTL analysis? I am surprised by the low number compared to the high number of mQTLs with >14k CpGs. What is the function of the CpGs when only a minority is influencing gene expression?

Influence of epilepsy status/environment: I disagree with the authors that the disease status or medication does not influence the results. The argument that these influences should be genotype independent is not supported by evidence. In fact, publications point towards genotype dependent epigenetic or transcriptional changes in response to environment or disease (GxE eQTL and GxE mQTL). For example: <http://genome.cshlp.org/content/early/2016/10/18/gr.209759.116>
<https://www.ncbi.nlm.nih.gov/pubmed/28530654>

As this is not a case control design, it is difficult to control for disease or medication specific effects, but it might be possible to use types of medication, dosage or duration of medication as proxy, similar to numbers of epileptic seizures, number of disease years etc.

The authors demonstrate temporal stability of hippocampal QTLs by comparison to other published hippocampus QTL datasets, I was wondering if this dataset <http://www.biorxiv.org/content/early/2017/07/07/142927> could also be used to compare eQTL and mQTL finding with PFC tissue and examine or cross-validate findings in a different brain region.

I did not access the provided website with the deposited data, however, a easily searchable interface with mQTL and eQTL data would be helpful for those who are not bioinformatics aficionados

I think the presented dataset is a great resource, similar to other resources that have been published before or are available online. However, although important, no molecular mechanism is shown and the analyses rely on correlational evidence. I fully agree with the authors that this might be beyond the scope of this paper but my worries from a molecular point of view is that the structure of the genome, it's function and the combination of multiple genome-wide datasets and their overlap will inevitably result in significant findings and it is difficult to derive causal relationships and mechanisms from that. An example would be that epilepsy induced gene transcription does induce DNAm changes, potentially in a genotype dependent fashion.

NCOMMS-17-05407A Response to the Reviewers

We are glad to hear that Reviewers #1 & #2 were satisfied by our revision of the manuscript. We appreciate the new comments and suggestions of Reviewer #3. Corresponding to the comments of Reviewer #3, we have revised the manuscript and have prepared a point-by-point response addressing the critical issues. We hope that the revised manuscript will be acceptable for publication in Nature Communications.

Reviewer #3 (Remarks to the Author):

Ad 1) eQTM analysis: Thank you for clarifying this point, however, I was rather interested on a genome wide model that takes genotype and methylation in relation to gene expression into account. Is the DNAm ~ gene expression relationship for the 34 genes based on previously identified mQTL genes? i.e. are the 73 cis related CpG also found in the mQTL analysis? I am surprised by the low number compared to the high number of mQTLs with >14k CpGs. What is the function of the CpGs when only a minority is influencing gene expression?

Ad 2: Influence of epilepsy status/environment: I disagree with the authors that the disease status or medication does not influence the results. The argument that these influences should be genotype independent is not supported by evidence. In fact, publications point towards genotype dependent epigenetic or transcriptional changes in response to environment or disease (GxE eQTL and GxE mQTL). For example:

<http://genome.cshlp.org/content/early/2016/10/18/gr.209759.116>

<https://www.ncbi.nlm.nih.gov/pubmed/28530654>

As this is not a case control design, it is difficult to control for disease or medication specific effects, but it might be possible to use types of medication, dosage or duration of medication as proxy, similar to numbers of epileptic seizures, number of disease years etc.

Ad 3) The authors demonstrate **temporal stability of hippocampal QTLs** by comparison to other published hippocampus QTL datasets, I was wondering if this dataset <http://www.biorxiv.org/content/early/2017/07/07/142927> could also be used to compare eQTL and mQTL finding with PFC tissue and examine or cross-validate findings in a different brain region.

Ad 4) I did not access the provided **website with the deposited data**, however, a easily searchable interface with mQTL and eQTL data would be helpful for those who are not bioinformatics aficionados.

Point-by-point response to the comments of Reviewer #3.

(Comments of Reviewer #3 are presented in *ITALICS*)

Ad 1) eQTM analysis: *Thank you for clarifying this point, however, I was rather interested on a genome wide model that takes genotype and methylation in relation to gene expression into account. Is the DNAm ~ gene expression relationship for the 34 genes based on previously identified mQTL genes? i.e. are the 73 cis related CpG also found in the mQTL analysis? I am surprised by the low number compared to the high number of mQTLs with >14k CpGs. What is the function of the CpGs when only a minority is influencing gene expression?*

Response: The present study aims to identify *cis-regulatory* meQTLs and eQTLs in human hippocampal biopsies. We did not carry out genome-wide trans-QTL analyses for reasons explained in the Methods (page #15, lines 595-598; Methods Section “Statistical quantitative trait loci analyses”), in particular insufficient power and high false positive rate due to cross-hybridization of some CpG probes. As described in the Methods section (page #15, lines 599-604, “Correlation analysis of mRNA expression and DNA methylation”), we have initially explored the correlation between individual gene expression levels (15k probe-sets) and the methylation states of regional CpGs within a *cis*-window of ± 500 kb around the mRNA expression probe. Subsequently, we explored whether CpG methylation-gene expression correlations correspond with meQTLs and eQTLs, respectively. According to our eQTM results provided in the Supplementary Data 4 (column #AF “meQTL_FDR”), 47 out of 73 eQTM-CpGs were also hippocampal *cis*-meQTL-CpGs at an FDR of 1%. We agree that the number of eQTM-genes ($n = 34$) discovered in the present study is low. Potential reasons are given in the Discussion (page #11, lines 416-427). Considering that we have investigated only gene-level mRNA transcription (15k 3'-mRNA probe-sets) instead of exon-level eQTL analyses, and that the 344k CpGs investigated in the present *cis*-QTL analyses represent only 1.2% of all CpG sites of the human genome, we decided to refrain from drawing general conclusions about the function of CpG methylation states on gene expression.

Revision: To address the point raised by the Reviewer, we have extended the Discussion and listed two main reasons that may explain the relative low number of eQTM-genes detected in the present study:

Page #11, lines 423-427, second paragraph: “Accordingly, the number of *cis*-eQTLs and methylation-expression correlations should be much higher at the exon-level relative to the gene-level. Considering that we have investigated only 1.2% of the 28 million CpG sites in the human genome, the present methylation-expression correlations likely reflect only a small proportion of the real number of methylation-driven gene expressions”.

Ad 2: Influence of epilepsy status/environment: *I disagree with the authors that the disease status or mediation does not influence the results. The argument that these influences should be genotype independent is not supported by evidence. In fact, publications point towards genotype dependent epigenetic or transcriptional changes in response to environment or disease (GxE eQTL and GxE mQTL). For example:*

<http://genome.cshlp.org/content/early/2016/10/18/gr.209759.116>

<https://www.ncbi.nlm.nih.gov/pubmed/28530654>

As this is not a case control design, it is difficult to control for disease or medication specific

effects, but it might be possible to use types of medication, dosage or duration of medication as proxy, similar to numbers of epileptic seizures, number of disease years etc.

Response: Reviewer #3 addresses an interesting aspect pointing towards interactions between genetic variation and environment in allele-specific expression (GxE ASE/eQTL). With reference to the GxE publication cited by the Reviewer, Knowles et al. (PMID:28530654) identified 35 GxE interactions at an FDR of 10% by examining the influence of 30 environmental factors in ASE of 8,795 genes derived from whole blood cells of 922 human individuals. Notably, this GxE study demonstrates that the identified 35 GxE interactions are not common and explain relatively small proportions of variance of gene expression.

Compared to the strong impact of SNP genotypes on the CpG methylation states and mRNA transcription levels of *cis*-acting hippocampal meQTLs and eQTLs (average explained variance: 28%, range per probe set: 11% - 85%), the potential shift of methylation and transcription levels induced by GxE should have only marginal effects on the genotypic correlation of SNPs with methylation and expression levels. As suggested by the reviewer, we have estimated the proportion of variance explained by TLE-related clinical factors, such as a) number of epileptic seizures, b) duration of epilepsy, c) types of antiepileptic medication, and d) therapy outcome after epilepsy surgery. Overall, the average estimated proportion of variance of CpG methylation and gene expression of hippocampal meQTLs and eQTLs was relatively small, varying between 0.4% to 1.1% (range per probe: 0.0% - 14.9%). These findings indicate that the epilepsy status does not exert a substantial effect in the majority of hippocampal meQTLs/eQTLs. Moreover, our ISV-adjusted QTL analyses emphasize that the identified hippocampal meQTLs/eQTLs are not adversely affected by relevant hidden confounding factors, such as environmental factors or disease states. In addition, we have discussed the issue that an up-regulation of the expression of genes induced by epileptogenic processes may increase the power to detect epilepsy-related eQTLs/meQTLs (Discussion, page #9, lines 315-336, second paragraph). Notably, the majority of hippocampal meQTLs/eQTLs detected in the present study has also been observed in brain tissue of individuals without epilepsy.

Revision: By regression analysis we do not find evidence that the epilepsy status substantially influences hippocampal *cis*-meQTLs/-eQTLs identified in the present study. As suggested by the reviewer, we have extended the Discussion to address this issue more in details and added the following paragraph (Discussion, page #9, lines 316-336, second paragraph):

Emerging evidence suggests that interactions between genetic variation and environmental factors may contribute to eQTLs and meQTLs (44,45). However, conditional allele-dependent shifts of mRNA transcription levels by gene-by-environment (GxE) interaction seem to affect only a small fraction (0.4%) of the investigated eQTL-genes and explain relatively small proportions of variance of gene expression (44). To explore the potential influence of the epilepsy state on the present hippocampal *cis*-meQTLs/eQTLs, we have estimated the proportion of variance of CpG methylation and gene expression attributable to TLE-related clinical factors (number of epileptic seizures, duration of epilepsy, type of antiepileptic medication, and therapy outcome after epilepsy surgery). Compared to the strong impact of SNP genotypes on *cis*-acting hippocampal meQTLs and eQTLs (FDR of 1%; average explained variance: 28%, range: 11% - 85%), the average proportion of variance of CpG methylation and gene expression explained by

the investigated TLE-related factors was relatively small varying between 0.4% to 1.1% (range per probe-set: 0.0% - 14.9%). Moreover, our ISV-adjusted QTL analyses did not reveal evidence that the identified hippocampal meQTLs/eQTLs may be substantially influenced by hidden epilepsy-related or environmental factors. Together, these findings suggest that the epilepsy state exerts marginal effects on CpG methylation and gene expression in the majority of hippocampal eQTLs and meQTLs identified in the present study. However, the epilepsy state or environmental factors may induce an up-regulation of the expression levels of at least some genes, possibly even in a genotype-dependent manner, which thereby may increase the power to detect epilepsy trait-related eQTLs.

Ad 3) *The authors demonstrate temporal stability of hippocampal QTLs by comparison to other published hippocampus QTL datasets, I was wondering if this dataset <http://www.biorxiv.org/content/early/2017/07/07/142927> could also be used to compare eQTL and mQTL finding with PFC tissue and examine or cross-validate findings in a different brain region.*

Response: We want to thank the Reviewer to direct our attention to the novel PFC-related meQTL/eQTL study published online without peer review on bioRxiv. Notably, we have also provided reference data of *cis*-regulatory meQTLs and eQTLs derived from PFC tissue in the Supplementary Data 1 (Refs #20/ PMID:26619357 & #21/PMID:26619358) and Supplementary Data 3 (GTE_x). Since the PFC-related xQTL study suggested by the Reviewer is not published yet as a peer-reviewed article, it must be expected that it will undergo changes during the review process. We therefore find it premature to implement xQTL-results in the present study.

Revision: None.

Ad 4) *I did not access the provided website with the deposited data, however, a easily searchable interface with mQTL and eQTL data would be helpful for those who are not bioinformatics aficionados.*

Response: The significant (FDR of 1%) hippocampal *cis*-meQTLs and -eQTLs of the LD-pruned SNP dataset (n = 536,041) are reported in the Supplementary Data 1 (66,970 significant SNP-CpG methylation associations) and Supplementary Data 3 (1,337 SNP-3'-RNA expression associations). Complementary, Supplementary Data 5 & 6 provide more than 50k meaningful QTL-SNPs (FDR of 1%, CADD score >5) based on the dataset of imputed SNPs (n = 3.3 million), including extensive annotations of regulatory genomic elements and an overlap with GWAS trait-associated SNPs ($P > 5.0 \times 10^{-8}$). Together, the Supplementary Data report all relevant QTL findings and are easy to access (Excel Tables). Upon request of the reviewers, we have made publically available the entire QTL results of all probe-sets and all imputed SNPs (n = 3.2 million). The implementation of a publically accessible online database that contains all QTL results and regular updates of the rapidly evolving genomic annotations and epigenomic profiles, as suggested by the reviewer, would require significant personal resources which we currently do not have. If such a database is not curated on a regular basis it will be rapidly outdated. We hope that this is understandable.

Revision: None.

REVIEWERS' COMMENTS:

Reviewer #3 (Remarks to the Author):

The authors addressed my remaining concerns. Thank you!